# Inhibition of ULK1/2 and KRAS[G12C] controls tumor growth in preclinical models of lung cancer

Phaedra C Ghazi[1,2], Kayla T O'Toole[1,2], Sanjana Srinivas Boggaram[1,2], Michael T Scherzer[1,2], Mark R Silvis[1,2], Yun Zhang[3], Madhumita Bogdan[4], Bryan D Smith[4], Guillermina Lozano[3], Daniel L Flynn[4], Eric L Snyder[1,2,5], Conan G Kinsey[1,2,6], Martin McMahon[1,2,7]*

[1]Department of Oncological Sciences, University of Utah, Salt Lake City, United States; [2]Huntsman Cancer Institute, University of Utah, Salt Lake City, United States; [3]Department of Genetics, The University of Texas MD Anderson Cancer Center, Houston, United States; [4]Deciphera Pharmaceuticals, St. Lawrence, United States; [5]Department of Pathology, University of Utah, Salt Lake City, United States; [6]Department of Internal Medicine, Division of Medical Oncology, University of Utah, Salt Lake City, United States; [7]Department of Dermatology, University of Utah, Salt Lake City, United States

*For correspondence:
martin.mcmahon@hci.utah.edu

**Abstract** Mutational activation of *KRAS* occurs commonly in lung carcinogenesis and, with the recent U.S. Food and Drug Administration approval of covalent inhibitors of KRAS[G12C] such as sotorasib or adagrasib, KRAS oncoproteins are important pharmacological targets in non-small cell lung cancer (NSCLC). However, not all KRAS[G12C]-driven NSCLCs respond to these inhibitors, and the emergence of drug resistance in those patients who do respond can be rapid and pleiotropic. Hence, based on a backbone of covalent inhibition of KRAS[G12C], efforts are underway to develop effective combination therapies. Here, we report that the inhibition of KRAS[G12C] signaling increases autophagy in KRAS[G12C]-expressing lung cancer cells. Moreover, the combination of DCC-3116, a selective ULK1/2 inhibitor, plus sotorasib displays cooperative/synergistic suppression of human KRAS[G12C]-driven lung cancer cell proliferation in vitro and superior tumor control in vivo. Additionally, in genetically engineered mouse models of KRAS[G12C]-driven NSCLC, inhibition of either KRAS[G12C] or ULK1/2 decreases tumor burden and increases mouse survival. Consequently, these data suggest that ULK1/2-mediated autophagy is a pharmacologically actionable cytoprotective stress response to inhibition of KRAS[G12C] in lung cancer.

## eLife assessment

This is a mechanistic study showing the effect of combining inhibition of autophagy (through ULK1/2) and KRAS (using sotorasib) on KRAS mutant NSCLC making the study **valuable** to cancer biologists and more broadly in a clinical setting. The evidence generated by GEM mouse models and cell lines is **solid** but could be further strengthened by increasing the mouse cohort size. This study holds translational relevance beyond NSCLC to other indications that carry KRAS mutations.

## Introduction

Lung adenocarcinoma (LUAD) is responsible for ~50,000 deaths/year in the United States. Among the most common genetic drivers of LUAD are mutations in *KRAS* that encode KRAS-4A and -4B

oncoproteins, in which the substitution of cysteine for glycine at codon 12 (KRAS$^{G12C}$) is the most common mutation and is associated with exposure to tobacco smoke (*Riely et al., 2008*; *Ahrendt et al., 2001*). For many years, RAS (H-, K-, or N) oncoproteins were considered undruggable until the seminal work of Ostrem et al., which demonstrated the feasibility of developing covalent inhibitors of KRAS$^{G12C}$ that bind into the Switch II pocket of KRAS$^{G12C}$-GDP and derivatize the reactive thiol of cysteine 12 (*Uprety and Adjei, 2020*; *Ostrem et al., 2013*; *Janes et al., 2018*).

We and others have previously reported that inhibition of RAS>RAF>MEK>ERK signaling in melanoma or pancreatic cancer cells elicits increased autophagy, an intracellular macromolecule and organelle recycling mechanism, through the LKB1>AMPK>ULK1 signaling axis (*Silvis et al., 2023*; *Truong et al., 2020*; *Kinsey et al., 2019*; *Bryant et al., 2019*). Liver kinase B1 (LKB1) activates AMP kinases in response to cellular nutrient changes to maintain homeostasis (*Li and Zhu, 2020*). However, induction of autophagy has also been reported in response to MEK inhibition in models of KRAS$^{G12D}$-driven lung cancer with loss of LKB1 expression (*Bhatt et al., 2023*), suggesting that LKB1 is dispensable for increases in autophagy in lung cancer. LKB1 silencing is frequently observed in patients with KRAS-driven lung cancer, and such patients exhibit decreased overall survival and are more refractory to current therapeutic treatments than in LKB1-proficient KRAS-driven lung cancers (*Shackelford et al., 2013*; *Sanchez-Cespedes, 2011*). With the advent of U.S. Food and Drug Administration (FDA)-approved covalent KRAS$^{G12C}$ inhibitors, it is imperative to investigate whether autophagy occurs following inhibition of KRAS$^{G12C}$ signaling, especially since MEK inhibitors offer little clinical benefit accompanied by toxicity to lung cancer patients (*Blumenschein et al., 2015*). This prompted us to explore whether treatment with sotorasib leads to an increase in autophagy in KRAS$^{G12C}$ mutant lung cancer cells with and without the expression of LKB1.

At present, the only FDA-approved therapeutics that inhibit autophagy are the 4-amino-quinolone lysosomal inhibitors chloroquine (CQ) and hydroxychloroquine (HCQ) (*Mohsen et al., 2022*). Originally approved for treating malaria, HCQ acts by inhibiting Toll-like receptors (*Takeda et al., 2003*). It was also reported that HCQ inhibits acidification of the lysosome, thereby inhibiting autolysosome activity, although the precise mechanism of this remains obscure (*Al-Bari, 2015*; *Seglen et al., 1979*; *Mauthe et al., 2018*). Clinically, high concentrations of HCQ are required to achieve modest inhibition of autophagy in patients, suggesting that the inhibitory potency of these compounds may limit clinical responses. Indeed, co-targeting lysosomal function in combination with RAS-driven signaling has recently been tested in RAS-driven cancers in several clinical trials (*Kinsey et al., 2019*; *Bryant et al., 2019*; *Yang et al., 2018*; clinicaltrials.gov: NCT04214418, NCT04132505, NCT04386057, NCT04145297). However, there is a lack of clinical investigation on targeting autophagy with selective autophagy inhibitors in patients.

Autophagy is a complex intracellular recycling process important for maintaining cellular homeostasis under stressed conditions. The genetics and biochemistry of autophagy, first revealed in yeast (*Glick et al., 2010*), indicate numerous points of regulation of which the initiation and formation of the autophagosome, as well as the fusion of the autophagosome to the lysosome, are of key importance (*Glick et al., 2010*; *Zachari and Ganley, 2017*). ULK1/2 are protein serine/threonine kinases that form a complex with FIP200, ATG13, and ATG101, which integrates upstream signals from mTORC1 and AMPK energy-sensing pathways (*Zachari and Ganley, 2017*). Upon activation, ULK1/2 phosphorylate a number of substrates, including ATG13 (at serine 318, pS318), to initiate the formation of autophagosomes (*Zachari and Ganley, 2017*). There remains interest in the development of selective inhibitors of autophagy for use in a number of disease indications (*Lee et al., 2023*). DCC-3116 is one such 'switch-control' ULK1/2 inhibitor that is currently in phase 1/2 clinical trials either as a monotherapy or in combination with inhibitors of KRAS$^{G12C}$, MEK1/2, or EGFR (NCT04892017 and NCT05957367).

Here, we test the antitumor effects of DCC-3116 against preclinical models of KRAS$^{G12C}$-driven lung cancer either alone or in combination with sotorasib. We observed that cultured KRAS$^{G12C}$-driven lung cancer cells increase autophagy in response to inhibition of KRAS$^{G12C}$>RAF>MEK>ERK signaling. In addition, combined inhibition of KRAS$^{G12C}$ plus ULK1/2 leads to decreased cell proliferation and tumor growth. Moreover, using genetically engineered mouse (GEM) models of either KRAS$^{G12C}$/TP53$^{R172H}$- or KRAS$^{G12C}$/LKB1$^{Null}$-driven lung cancer (*Zafra et al., 2020*), we demonstrate that LKB1 is dispensable for increases in autophagy following KRAS$^{G12C}$ inhibition. Furthermore, LKB1 silencing diminishes the sensitivity of KRAS$^{G12C}$/LKB1$^{Null}$-driven lung cancer to KRAS$^{G12C}$ inhibition perhaps through the emergence of mixed adenosquamous cell carcinomas and mucinous adenocarcinomas. Adenosquamous

carcinomas (ASC) are observed in up to 4% of lung cancers, and the emergence of these tumors is observed as a resistance mechanism to current therapeutics in lung cancer patients (*Zhu et al., 2018*). Patients with ASC tend to exhibit substantially decreased overall survival compared to patients with adenocarcinomas or squamous carcinomas (*Li and Lu, 2018*). These data suggest that KRAS[G12C] mutant lung tumors with adenosquamous pathology may not respond to KRAS[G12C] or ULK inhibition. Clinical acquired resistance to sotorasib and adagrasib is emerging in KRAS[G12C] mutant lung cancer patients (*Canon et al., 2019*; *Fell, 2020*; *Hallin et al., 2020*; *Awad, 2021*; *Zhao et al., 2021*; *Blaquier et al., 2021*; *Tanaka et al., 2021*). Here, we demonstrate that KRAS[G12C]/LKB1[Null]-driven lung cancer cells that acquire resistance to sotorasib increase expression of RAS>RAF>MEK>ERK signaling, and no longer increase autophagy after sotorasib treatment. These cells do induce autophagy after MEK1/2 inhibition and decrease cellular proliferation and viability after treatment with trametinib, suggesting that sotorasib-resistant (SR) cells are still vulnerable to downstream KRAS[G12C] signaling inhibition.

## Results

### Human KRAS[G12C]-driven lung cancer cells are sensitive to co-inhibition of KRAS[G12C] and ULK1/2

To determine whether direct inhibition of KRAS[G12C] in lung cancer cells influences autophagy, human KRAS[G12X]-driven lung cancer cell lines were engineered to express a chimeric fluorescent autophagy reporter (FAR) comprised of mCherry fused to EGFP fused to LC3 (*Kinsey et al., 2019*; *Kimura et al., 2007*). LC3 targets the chimeric FAR to the autophagosome where the pH-sensitive EGFP moiety is quenched in the acidic environment of the autolysosome (pH < 5). Hence, an increase in the ratio of mCherry:EGFP fluorescence is indicative of increased autophagy (*Kinsey et al., 2019*; *Kimura et al., 2007*). All KRAS[G12C]-driven lung cancer cells displayed dose-dependent increases in autophagy following sotorasib treatment as indicated by an increase in the mCherry:EGFP ratio detected in live cells (*Figure 1A–C*, *Figure 1—figure supplement 1A–D*). Since sotorasib is reported to bind to >300 proteins, we ruled out nonspecific effects of sotorasib on autophagy by treating a KRAS[G12V]-driven NSCLC cell line (COR-L23) with sotorasib, which failed to show any change in mCherry:EGFP fluorescence (*Figure 1—figure supplement 1E*; *Wang et al., 2023*). Moreover, as predicted by previous experiments, all KRAS[G12X]-driven NSCLC cell lines displayed a dose-dependent increase in autophagy following treatment with trametinib, a MEK1/2 inhibitor (*Figure 1—figure supplement 1F–J*; *Kinsey et al., 2019*).

Next, we tested the effects of DCC-3116 on autophagy in KRAS[G12C]-driven NSCLC cell lines. As anticipated, DCC-3116 inhibited both basal and sotorasib-induced autophagy in all KRAS[G12C]-driven cell lines as assessed using the FAR (*Figure 1A–C*). In addition, DCC-3116 treatment, either alone or in combination with sotorasib, led to decreased pS318-ATG13, one of ULK1's immediate downstream substrate (*Figure 1—figure supplement 2A–C*). To test whether DCC-3116 might show additive or synergistic antiproliferative effects when combined with sotorasib, human KRAS[G12C]-driven NSCLC cells were treated with varying concentrations of both drugs for 72 hr with cell proliferation assessed using a cell viability assay. Interestingly, we observed synergistic antiproliferative effects of DCC-3116 plus sotorasib in NCI-H2122 and Calu-1 cell lines (*Figure 1G and H*), but not in NCI-H358 cells, which are noted to be exquisitely sensitive to single-agent sotorasib (*Figure 1I*). Using the IncuCyte platform, we measured cell proliferation over time and noted that the combination of sotorasib plus DCC-3116 was superior to either single agent in NCI-H2122 and Calu-1 cells (*Figure 1D and E*, *Figure 1—figure supplement 2D and E*), but not in NCI-H358 cells (*Figure 1F*, *Figure 1—figure supplement 2F*). Intriguingly, we observed consistent inhibition of NCI-H358 cell proliferation with single-agent DCC-3116 (*Figure 1F*, *Figure 1—figure supplement 2F*).

We next proceeded to test the antitumor effects of DCC-3116, either alone or in combination with sotorasib, against xenografted tumors derived from injection of NCI-H2122, Calu-1, or NCI-H358 cells into immunocompromised mice. Here, we observed clear cooperation of sotorasib plus DCC-3116 in tumors derived from NCI-H2122 and Calu-1 cells (*Figure 1J and K*) but not from NCI-H358 cells, which are noted for their sensitivity to low concentrations of sotorasib (*Figure 1L*). To confirm the inhibitory activity of sotorasib in the cell lines used, cell extracts of control vs. sotorasib-treated NSCLC cells were immunoblotted for phospho-ERK1/2. As expected, pERK1/2 was decreased in all cell lines following 2 hr of sotorasib treatment (*Figure 1—figure supplement 2G–I*). In addition,

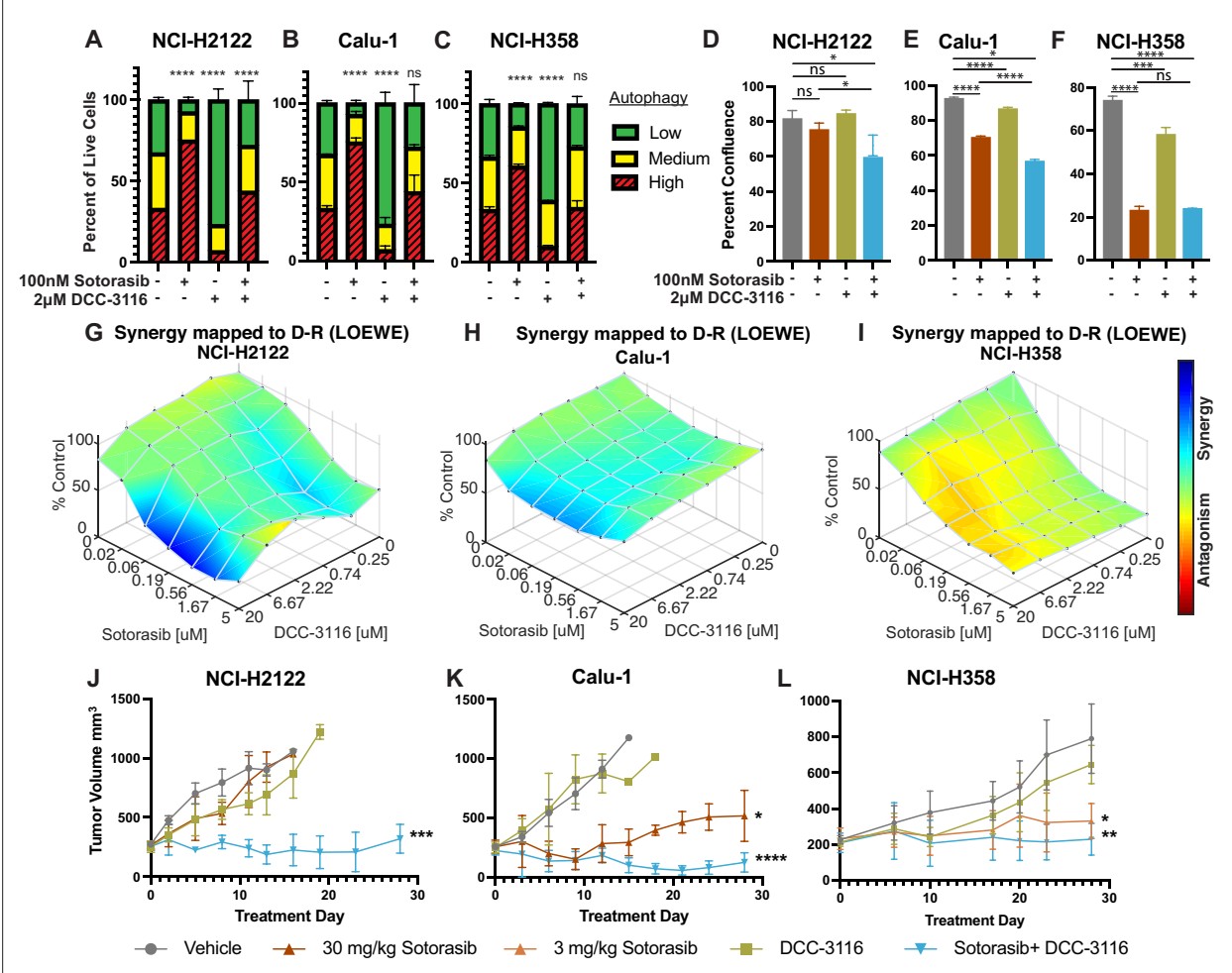

**Figure 1.** Human KRAS[G12C]-driven lung cancer cells are sensitive to co-inhibition of KRAS[G12C] and ULK1/2. (**A–C**) Human KRAS[G12C]-driven cell lines NCI-H2122 (**A**), Calu-1 (**B**), and NCI-H358 (**C**) increase autophagy as assessed by mCherry-EGFP-LC3 reporter after 48 hr of sotorasib treatment and decrease autophagy after 48 hr of DCC-3116 treatment. Red = high autophagy, yellow = medium autophagy, green = low autophagy. Statistical significance was determined by comparing autophagy levels to DMSO control, and an ordinary one-way ANOVA with Dunnett's multiple comparisons was used. Ns = not significant, *p<0.05, **p<0.01, ***p<0.001, ****p<0.0001. N = 9 biological replicates. (**D–F**) Quantification of percent confluence of human KRAS[G12C]-driven cell lines at 72 hr post-drug treatment. Statistical significance was determined by an ordinary one-way ANOVA. Ns = not significant, *p<0.05, **p<0.01, ***p<0.001, ****p<0.0001. N = 3 biological replicates. (**G–I**) In vitro synergy assay of human KRAS[G12C]-driven cell lines using the Loewe method after 72 hr of treatment. N = 3 biological replicates. (**J–L**) Tumor volume measured over 28 days of treatment in mice inoculated with NCI-H2122 (**J**), Calu-1 (**K**), and NCI-H358 (**L**) cells. Vehicle and sotorasib were administered once daily via oral gavage and DCC-3116 was formulated in the chow. Statistical significance was determined by an ordinary one-way ANOVA compared to vehicle-treated tumors. Ns = not significant, *p<0.05, **p<0.01, ***p<0.001, ****p<0.0001. N = 4–5 mice per treatment.

The online version of this article includes the following source data and figure supplement(s) for figure 1:

**Source data 1.** Uncropped and labeled gels for *Figure 1*.

**Source data 2.** Raw unedited gels for *Figure 1*.

**Figure supplement 1.** Human KRAS-driven lung cancer cells increase autophagy after KRAS[G12C] or MEK inhibition.

**Figure supplement 1—source data 1.** Uncropped and labeled gels for *Figure 1—figure supplement 1*.

**Figure supplement 1—source data 2.** Raw unedited gels for *Figure 1—figure supplement 1*.

**Figure supplement 2.** Human KRAS[G12C]-driven lung cancer cell lines increase autophagy and decrease cellular proliferation after sotorasib treatment.

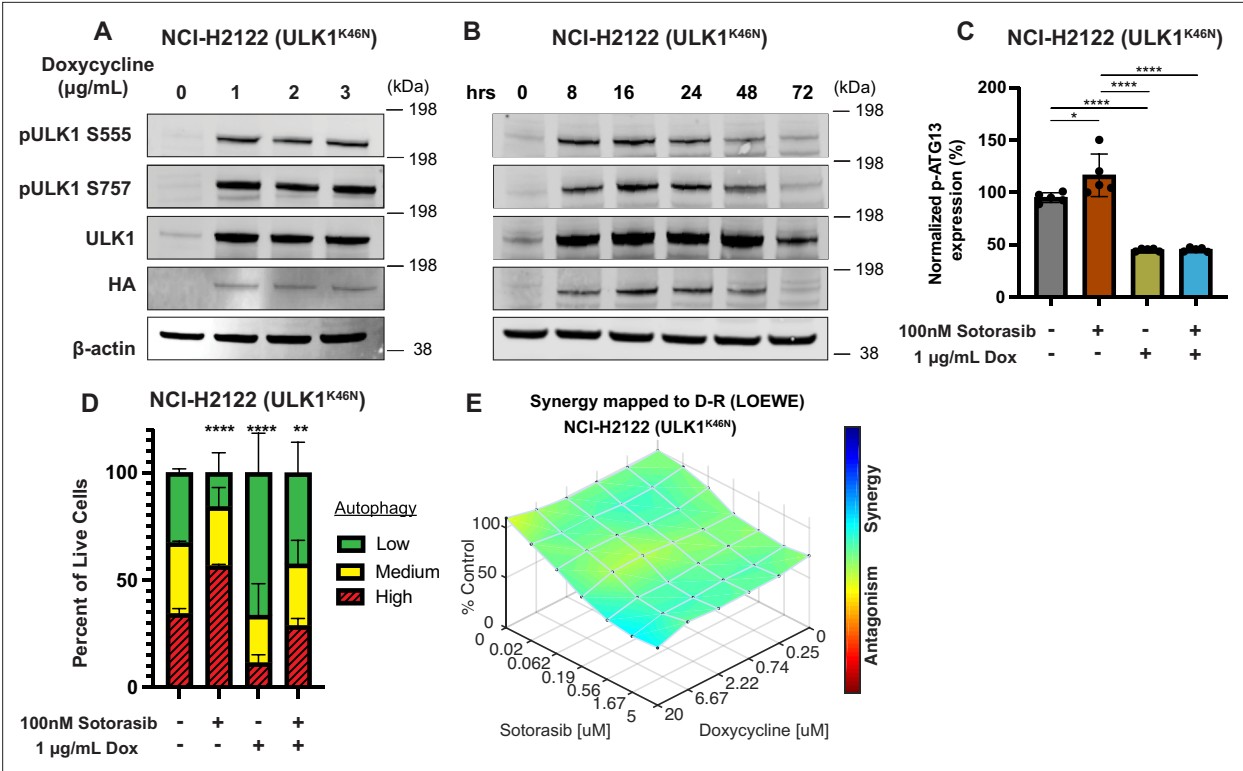

**Figure 2.** Genetic inhibition of ULK1 decreases autophagy and cooperates with sotorasib to reduce cell viability. (**A**) Immunoblot of NCI-H2122:ULK1[K46N] cells after 24 hr of doxycycline treatment. (**B**) Immunoblot of NCI-H2122:ULK1[K46N] cells treated with 1 ug/mL doxycycline over time (hours). (**C**) ELISA of pS318-ATG13 expression after 16 hr of doxycycline treatment. Statistical significance was determined by an ordinary one-way ANOVA. Ns = not significant, *$p<0.05$, **$p<0.01$, ***$p<0.001$, ****$p<0.0001$. N = 3 biological replicates. (**D**) NCI-H2122:ULK1[K46N] cells were engineered to express the mCherry-EGFP-LC3 reporter, and a decrease in autophagy was demonstrated after 48 hr of doxycycline treatment. N = 3. All statistical significance was measured using an ordinary one-way ANOVA with Dunnett's multiple comparisons test. *$p<0.05$, **$p<0.01$, ***$p<0.001$, ****$p<0.0001$. (**E**) In vitro synergy assay of NCI-H2122:ULK1[K46N] cells treated with DMSO control, sotorasib, and/or doxycycline over 48 hr using the Loewe method. N = 3 biological replicates.

The online version of this article includes the following source data for figure 2:

**Source data 1.** Uncropped and labelled gels for *Figure 2*.

**Source data 2.** Raw unedited gels for *Figure 2*.

we detected by immunoblotting the expected electrophoretic mobility shift of KRAS[G12C], which is indicative of covalent binding of sotorasib (561 daltons) to cysteine 12 of KRAS[G12C] (*Figure 1—figure supplement 2G–I*; *Canon et al., 2019*).

## Genetic inhibition of ULK1 decreases autophagy and cooperates with sotorasib to reduce cell viability

As an alternate way to assess the role of ULK1/2 in sotorasib-induced autophagy, we conditionally expressed a kinase-inactive, dominant negative form of ULK1 (ULK1[K46N]) under the control of a tetracycline transactivator (Tet[ON]) (*Hara et al., 2008*). Doxycycline treatment of NCI-H2122/Tet[ON]::ULK1[K46N] cells led to a substantial increase in total ULK1 expression within 8 hr, which could be inferred to also be phosphorylated at known sites (pS555 and pS747) of regulatory phosphorylation of ULK1 (*Figure 2A and B*). As anticipated, expression of ULK1[K46N] led to decreased phosphorylation of pS318-ATG13 in a manner comparable to treatment with DCC-3116 (*Figure 2C*). In addition, expression of ULK1[K46N] led to a decrease in both basal and sotorasib-induced autophagy as assessed using the FAR. Finally, expression of ULK1[K46N] displayed synergistic antiproliferative effects in NCI-H2122

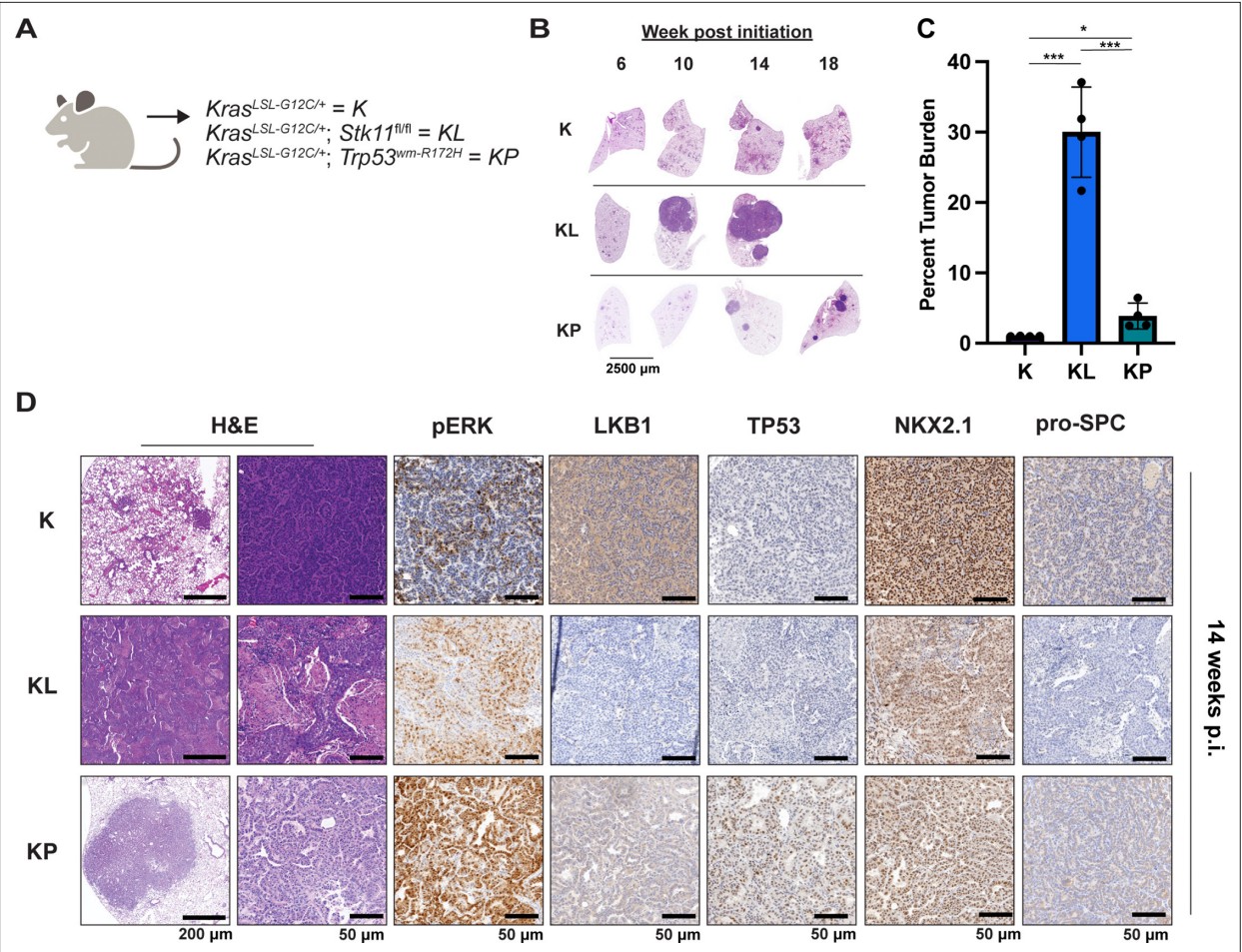

**Figure 3.** Either LKB1 silencing or expression of dominant-negative TP53[R172H] cooperates with KRAS[G12C] in genetically engineered mouse (GEM) models of lung cancer. (**A**) Schematic of genotypes of GEM models and abbreviations. Panel (**A**) was created with BioRender.com and published using a CC BY-NC-ND license with permission. (**B**) Representative images of lung lobes from GEM models at indicated time points post-initiation of lung tumorigenesis. (**C**) Quantification of lung tumor burdens from GEM models 14 weeks post-initiation of tumorigenesis. Statistical analysis was performed using an ordinary one-way ANOVA. Ns = not significant, *p<0.05, **p<0.01, ***p<0.001, ****p<0.0001. N = 4 mice. (**D**) Hematoxylin and eosin (H&E) and immunohistochemical analysis of representative lung sections from GEM models 14 weeks post-initiation of tumorigenesis. p.i. = post-initiation.

The online version of this article includes the following figure supplement(s) for figure 3:

**Figure supplement 1.** Loss of LKB1 expression leads to mixed adenosquamous cell carcinoma (ASC) and mucinous adenocarcinoma (ADC) lung tumors in KRAS[G12C]-driven genetically engineered mouse (GEM) models.

cells when combined with sotorasib for 48 hr, suggesting that both genetic and pharmacological inhibition of ULK1 signaling had similar effects on NCI-H2122 cells (*Figure 2E*).

## Either LKB1 silencing or expression of dominant-negative TP53[R172H] cooperates with KRAS[G12C]-in GEM models of lung cancer

To complement the use of cultured human lung cancer cells and further test how additional genetic alterations influence KRAS[G12C]-driven lung tumor progression and treatment response, we generated genetically engineered mouse (GEM) models of KRAS[G12C]-driven lung cancer. To that end, we crossed *Kras[LSL-G12C/+]* (K) (*Zafra et al., 2020*) mice to either (1) *Stk11[fl/fl]* (*Nakada et al., 2010*) mice to generate KL mice or (2) *Trp53[wm-R172H]* to generate KP mice (*Lang et al., 2004*). Intranasal administration of an adenovirus encoding CRE recombinase under the control of a CMV promoter (Ad-CMV-CRE) to KL mice initiates expression of KRAS[G12C] plus silencing of LKB1 in alveolar type 2 (AT2) cells. Similar treatment of KP mice initiates expression of KRAS[G12C] plus dominant-negative TP53[R172H] in AT2 cells

(*Figure 3A*, *Figure 3—figure supplement 1A*, Key resources table). At euthanasia, 14 weeks post-initiation (p.i.), compared to *Kras*<sup>LSL-G12C/+</sup> (K) mice, KL and KP mice displayed both accelerated lung cancer formation (*Figure 3B*) and increased tumor burden (*Figure 3C*). Histopathological analysis of H&E-stained sections of KRAS<sup>G12C</sup> only-driven lung tumors, also at 14 weeks p.i., indicated that they comprised mainly of atypical adenomatous hyperplasia (AAH) and low-grade small LUADs. KRAS<sup>G12C</sup>/LKB1<sup>Null</sup>-driven lung tumors comprised AHH, adenocarcinomas, ASC, and mucinous adenocarcinomas. Finally, KRAS<sup>G12C</sup>/TP53<sup>R172H</sup>-driven lung tumors consisted of AAH and a mix of low- and higher-grade LUADs (*Figure 3—figure supplement 1B*). Immunohistochemical analysis of tumor sections indicated that phospho-ERK1/2 (pERK1/2) was detected in KRAS<sup>G12C</sup>-driven lung tumors regardless of cooperating alterations in LKB1 or TP53 (*Figure 3D*). However, the intensity of pERK1/2 staining across all three GEM models, and even between lesions in the same lung lobe (*Figure 3—figure supplement 1C*) displayed substantial variability. Moreover, phospho-AKT1-3 (a surrogate for activation of PI3'-kinase signaling) was low across all KRAS<sup>G12C</sup>-driven lung tumors, although some individual cells in tumors had readily detected pAKT1-3 (*Figure 3—figure supplement 1D*). Similar analysis confirmed the absence of LKB1 in KRAS<sup>G12C</sup>/LKB1<sup>Null</sup>-driven lung tumors, but its presence in KRAS<sup>G12C</sup> only and KRAS<sup>G12C</sup>/TP53<sup>R172H</sup>-driven lung tumors. Similarly, KRAS<sup>G12C</sup>/TP53<sup>R172H</sup>-driven lung tumors displayed elevated expression of TP53 that was not observed in KRAS<sup>G12C</sup> only or in KRAS<sup>G12C</sup>/LKB1<sup>Null</sup>-driven lung tumors (*Figure 3D*). Previously published data demonstrated that KRAS-driven lung cancer cells occasionally suppress expression of NK2 homeobox 2 (NKX2.1), a master-regulatory transcription factor required for normal lung development (*Moisés et al., 2017*). In addition, KRAS-driven but NKX2.1-deficient lung tumors display a mucinous adenocarcinoma pathology and other transcriptional features of a pulmonary-to-gastric lineage switch (*Snyder et al., 2013*). Although mucinous adenocarcinomas were detected in lung sections from KL mice (*Figure 3D*), these KRAS<sup>G12C</sup>/LKB1<sup>Null</sup> tumor cells retained expression of NKX2.1 (*Figure 3D*, *Figure 3—figure supplement 1E*), and some cells also expressed the gastrointestinal transcription factor HNF4α (*Figure 3—figure supplement 1E*). Finally, we assessed expression of the AT2-specific pro-surfactant protein C (SPC), a surfactant important for the function of normal lung (*Mason and Dobbs, 2016*), and noted its uniform expression in KRAS<sup>G12C</sup>-only-driven lung tumors, but more variable expression in KRAS<sup>G12C</sup>/LKB1<sup>Null</sup>- and KRAS<sup>G12C</sup>/TP53<sup>R172H</sup>-driven lung tumors (*Figure 3D*).

## Key resources table

| Reagent type (species) or resource | Designation | Source or reference | Identifiers | Additional information |
|---|---|---|---|---|
| Genetic reagent (*Mus musculus*) | *Kras*<sup>LSL-G12C</sup> | The Jackson Laboratory; PMID:32792368 | Strain #:033068 | Dow Lab |
| Genetic reagent (*M. musculus*) | *Stk11*<sup>fl/fl</sup> (mouse gene is *Stk11* and protein is referred to as LKB1) | The Jackson Laboratory; PMID:21124450 | Strain #:014143 | Morrison Lab |
| Genetic reagent (*M. musculus*) | *Trp53*<sup>wm-R172H</sup> | PMID:30262850 | | Lozano Lab |
| Genetic reagent (*M. musculus*) | Nod.Cg-Prkdcscid/J | The Jackson Laboratory | Strain #:001303 | |
| Strain (adenovirus) | Ad5-CMV-CRE | University of Iowa Viral Vector Core | VVC-U of Iowa-5 | |
| Cell line (*Homo sapiens*) | 293T | ATCC | CRL-3216 | |
| Cell line (*H. sapiens*) | NCI-H2122 | ATCC | CRL-5985 | |
| Cell line (*H. sapiens*) | Calu-1 | ATCC | HTB-54 | |
| Cell line (*H. sapiens*) | NCI-H358 | ATCC | CRL-5807 | |
| Cell line (*H. sapiens*) | NCI-H23 | ATCC | CRL-5800 | |
| Cell line (*H. sapiens*) | Cor-L23 | Sigma-Aldrich | 92031919 | |
| Cell line (*M. musculus*) | KL70 | Tumor-derived cell line | | |
| Antibody | Anti-β-actin (mouse monoclonal) | Cell Signaling Technology | Cat# 3700 | WB (1:10,000) |

*Continued on next page*

*Continued*

| Reagent type (species) or resource | Designation | Source or reference | Identifiers | Additional information |
|---|---|---|---|---|
| Antibody | Anti-phospho-AKT S473 (rabbit monoclonal) | Cell Signaling Technology | Cat# 4060 | WB (1:1000) IHC (1:100) |
| Antibody | Anti-Akt (mouse monoclonal) | Cell Signaling Technology | Cat# 2920 | WB (1:1000) |
| Antibody | Anti-phospho AMPKa S485 (rabbit monoclonal) | Cell Signaling Technology | Cat# 2537 | WB (1:1000) |
| Antibody | Anti-phospho AMPKa T172 (rabbit monoclonal) | Cell Signaling Technology | Cat# 2535 | WB (1:1000) |
| Antibody | Anti-AMPK (rabbit monoclonal) | Cell Signaling Technology | Cat# 5831 | WB (1:1000) |
| Antibody | Anti-phospho-ATG13 S318 (rabbit polyclonal) | Rockland Immunochemicals | Cat# 600-401C49 | ELISA (1:500) |
| Antibody | Anti-ATG13 (rabbit monoclonal) | Cell Signaling Technology | Cat# 13272 | ELISA (1:500) |
| Antibody | Anti-HNF4α (rabbit monoclonal) | Cell Signaling Technology | Cat# 3113 | IHC (1:500) |
| Antibody | Anti-LKB1 (IHC formulated) (rabbit monoclonal) | Cell Signaling Technology | Cat# 13031 | IHC (1:250) |
| Antibody | Anti-LKB1 (rabbit monoclonal) | Cell Signaling Technology | Cat# 3050 | WB (1:1000) |
| Antibody | Anti-NKX2.1 (rabbit monoclonal) | Abcam | Cat# ab76013 | IHC (1:2000) |
| Antibody | Anti-p44/p42 ERK1/2 (mouse monoclonal) | Cell Signaling Technology | Cat# 4696 | WB (1:1000) |
| Antibody | Anti-phospho-p44/42 Thr202/Tyr204 ERK1/2 (rabbit monoclonal) | Cell Signaling Technology | Cat# 4377 | WB (1:1000) IHC (1:600) |
| Antibody | Anti-TP53 (rabbit polyclonal) | Leica Biosystems | NCL-L-p53-CM5p | IHC (1:1000) WB (1:2,000) |
| Antibody | Anti-Pro-Surfactant Protein C (pSPC) (rabbit monoclonal) | MilliporeSigma | Cat# AB3786 | IHC (1:2000) |
| Antibody | Anti-RAS (rabbit monoclonal) | Cell Signaling Technology | Cat#3965 | WB (1:1000) |
| Antibody | Streptavidin-Poly-HRP antibody | Thermo Fisher | Cat# 21140 | ELISA (1:4000) |
| Antibody | Anti-phospho-ULK1 S555 (rabbit monoclonal) | Cell Signaling Technology | Cat# 5869 | WB (1:500) |
| Antibody | Phospho-ULK1 S757 (rabbit monoclonal) | Cell Signaling Technology | Cat# 14202 | WB (1:500) |
| Antibody | ULK1 (rabbit monoclonal) | Cell Signaling Technology | Cat# 8054 | WB (1:500) |
| Antibody | IRDye 800 CW Goat anti-Rabbit IgG | LI-COR | Cat# 926-32211 | WB (1:20,000) |
| Antibody | IRDye 680LT Donkey anti-Mouse IgG | LI-COR | Cat# 926-68022 | WB (1:20,000) |
| Recombinant DNA reagent | pCW57-MCS1-P2A-MSC2 (Blast) | Addgene | Plasmid # 80921 | |
| Recombinant DNA reagent | psPAX2 | Addgene | Plasmid # 12260 | |
| Recombinant DNA reagent | pMD2.G | Addgene | Plasmid # 12259 | |
| Recombinant DNA reagent | pLV-ULK1$^{K46N}$ | Vector Builder | | |
| Recombinant DNA reagent | pBabePuro:mCherry-GFP-LC3 | Addgene | Plasmid # 22418 | |
| Recombinant DNA reagent | pUltra-Hot | Addgene | Plasmid # 24130 | |
| Chemical compound, drug | Sotorasib | Deciphera Pharmaceuticals | | |
| Chemical compound, drug | DCC-3116 | Deciphera Pharmaceuticals | | |
| Chemical compound, drug | DCC-3116 formulated chow | Deciphera Pharmaceuticals, Research Diets | | |
| Chemical compound, drug | Trametinib | Shanghai Biochem Partner | Cat# BCP02307 | |
| Chemical compound, drug | Corn oil | Sigma-Aldrich | Cat# C8267 | |
| Chemical compound, drug | DMSO | Sigma-Aldrich | Cat# D8418 | |

# Antitumor effects of DCC-3116 and sotorasib either alone or in combination in GEM models of KRAS$^{G12C}$-driven lung cancer

The importance of autophagy in the initiation and/or progression of KRAS- or BRAF-driven lung cancer was revealed through analysis of GEM models in which deletion of key autophagy genes (Atg5 or Atg7) accompanied the initiation of oncoprotein expression (*Guo et al., 2013*; *Bhatt et al., 2019*; *Khayati et al., 2020*; *Poillet-Perez et al., 2018*; *Rao et al., 2014*). Hence, we tested whether DCC-3116-mediated inhibition of ULK1/2, starting at tumor initiation in KL mice, would inhibit tumor growth. Therefore, KL mice were initiated with Ad-CMV-CRE and then, starting 2 days p.i., treated with either vehicle or DCC-3116 (using a chow formulation) for 12 weeks (*Figure 4A*) at which time mice were euthanized for analysis. DCC-3116-treated KL mice displayed significantly decreased tumor burden compared to control KL mice (*Figure 4B and C*).

To test whether combined inhibition of KRAS$^{G12C}$ plus ULK1/2 would inhibit growth of established KRAS$^{G12C}$/LKB1$^{Null}$ lung tumors, tumor-bearing KL mice at 10 weeks p.i. were administered sotorasib or DCC-3116 either alone or in combination for 8 weeks or until the mice met predetermined euthanasia endpoints (*Figure 4D*). Compared to vehicle, all three treatments significantly increased the survival of tumor-bearing KL mice (*Figure 4E*). Although there was a trend toward improved survival of KL mice treated with the combination of sotorasib plus DCC-3116, these differences were not statistically significant (*Figure 4E*). We also utilized micro-computed tomography (microCT) to monitor the response of tumors (*Johnson, 2007*) that were large enough to be detected throughout drug treatment (*Figure 4—figure supplement 1A*). As expected, lung tumors in vehicle-treated mice rapidly increased in size (*Figure 4—figure supplement 1A*). The growth of KRAS$^{G12C}$/LKB1$^{Null}$ tumors in mice treated with sotorasib, DCC-3116, or the combination was delayed compared to those in vehicle-treated mice, but tumors ultimately progressed on treatment (*Figure 4—figure supplement 1A*). However, some lung tumors in mice treated with the sotorasib plus DCC-3116 drug combination did not appear to change their size over the course of treatment (*Figure 4—figure supplement 1A*). Throughout the course of treatment, we observed substantial ($\geq$20%) weight loss in four out of five sotorasib-only treated mice and one combination-treated mouse that reached our euthanasia criteria (*Figure 4—figure supplement 1B*). Following euthanasia, we performed immunohistochemical analysis on lung sections from control vs. drug-treated mice. In KL mice treated with sotorasib, either alone or in combination with DCC-3116, we noted a predominance of ASC and mucinous adenocarcinomas at the expense of AAH and adenocarcinoma (*Figure 4—figure supplement 1C*). These results suggest that KRAS$^{G12C}$/LKB1$^{Null}$-driven ASC or mucinous adenocarcinomas may be less sensitive to sotorasib treatment (*Figure 4—figure supplement 1A–C*). Intriguingly, single-agent DCC-3116 treatment also slowed the growth of established tumors compared to vehicle treatment (*Figure 4—figure supplement 1A*), but this did not inhibit the growth of AAH or adenocarcinomas (*Figure 4—figure supplement 1C*). Immunohistochemistry revealed the expected decrease in pERK1/2 and pAKT1-3 staining in tumors treated with sotorasib, either alone or in combination with DCC-3116, compared to mice treated with vehicle or single-agent DCC-3116 alone (*Figure 4G–I*). NKX2.1 expression was detected in all lung tumors regardless of the treatment group (*Figure 4G*). Finally, many mice, regardless of the treatment type, developed lung tumors with the aforementioned mucinous adenocarcinoma phenotype, expressing HNF4α and stained positive for various mucins (Alcian-Blue/PAS-positive) (*Figure 4G*).

To further compare and contrast the effects of sotorasib and DCC-3116, either alone or in combination in GEM models of lung cancer and because of noted correlations between loss of LKB1 and insensitivity of cancers to various types of therapy, we employed our KP mouse model (*Shackelford et al., 2013*; *Pons-Tostivint et al., 2021*). KRAS$^{G12C}$/TP53$^{R172H}$-driven lung tumorigenesis was initiated in adult KP mice and at 10 weeks p.i. tumor-bearing mice were treated with (1) vehicle control, (2) sotorasib, (3) DCC-3116, or (4) combination of sotorasib plus DCC-3116 for 4 weeks (*Figure 5A*) at which time mice were euthanized for analysis as described above. Interestingly, compared to control, there was a significant reduction in lung tumor burden in response to all three treatments (*Figure 5B*). In addition, in all sotorasib-treated mice, either alone or in combination with DCC-3116, regression of preexisting lung tumors was detected by microCT scanning (*Figure 5C*, *Figure 5—figure supplement 1*). In mice treated with single-agent DCC-3116, although an overall reduction in tumor burden was observed, some tumors detectable by microCT scanning continued to increase in size throughout treatment (*Figure 5B*, *Figure 5—figure supplement 1*). Moreover, in this experiment, we noted no

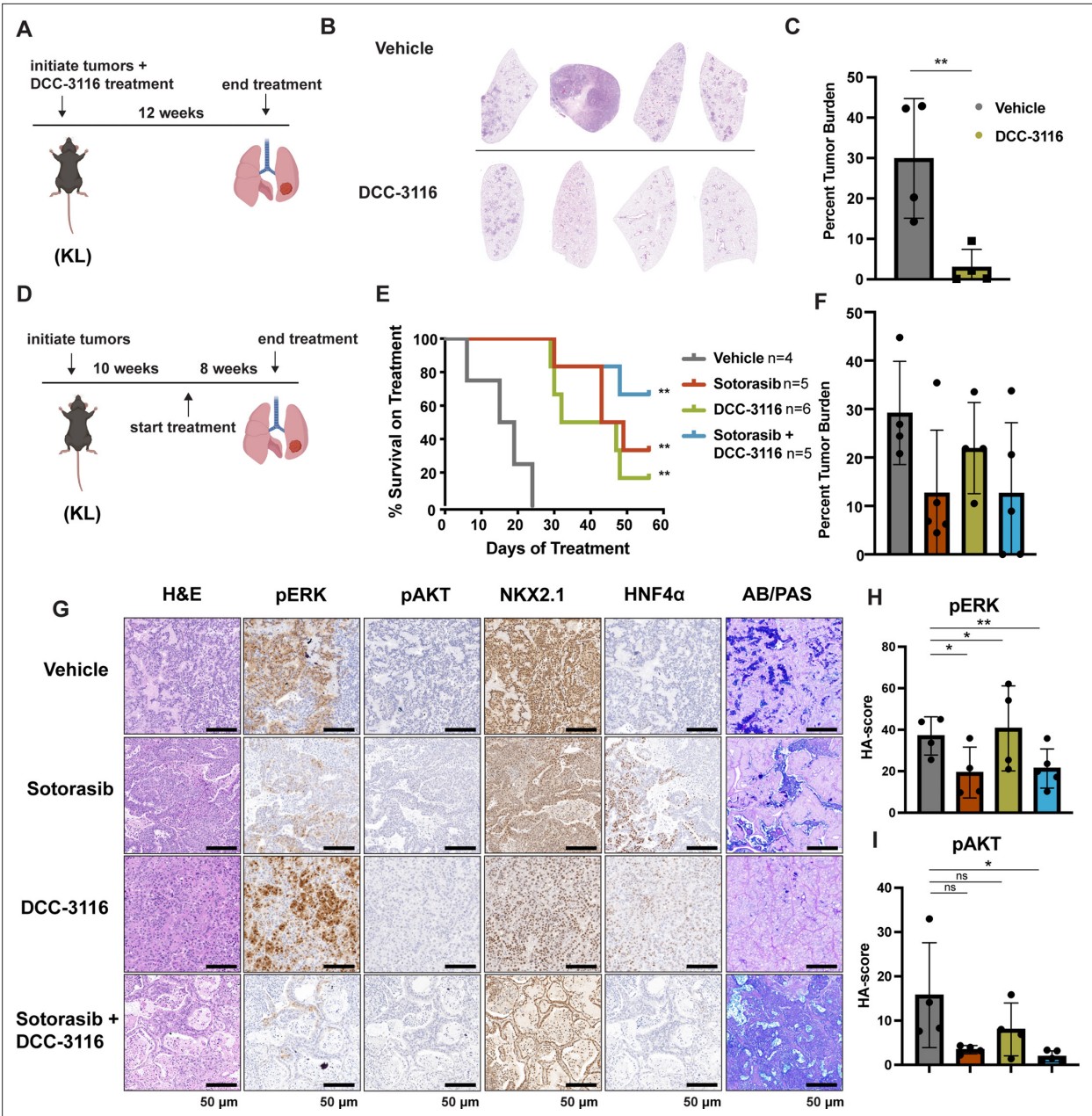

**Figure 4.** Combined inhibition of KRAS[G12C] and ULK1/2 decreases tumor initiation and increases survival in KL genetically engineered mouse (GEM) models. (**A**) Schematic of lung tumor prevention dosing strategy of KL GEMs. Panel (**A**) was created with BioRender.com and published using a CC BY-NC-ND license with permission. DCC-3116 was administered in drug-formulated chow. N = 4–6 mice. (**B**) Representative images of lung lobes from GEM models 12 weeks post-initiation of tumorigenesis and DCC-3116 treatment. (**C**) Quantification of tumor burden of (**B**). Statistical analysis was measured by an unpaired Student's *t*-test. Ns = not significant, *p<0.05, **p<0.01, ***p<0.001, ****p<0.0001. N = 4 mice. (**D**) Schematic of treating tumor-bearing KL mice with vehicle control, 30 mg/kg sotorasib, chow containing DCC-3116 or the combination. Panel (**D**) was created with BioRender.com and published using a CC BY-NC-ND license with permission. DCC-3116 was administered in drug-formulated chow (Key resources table). Mice were treated daily for 56 days or until termination criteria were reached, whichever was reached first. N = 4–6 mice. (**E**) Kaplan–Meier survival curve of survival on treatment of KL mice treated as indicated. Statistical analysis was performed using a log-rank test. **p<0.01, N = 4–6 mice. (**F**) Quantification of tumor burden of (**E**). (**G**) H&E analysis of representative lung sections from KL mice after treatment. AB/PAS = Alcian Blue Periodic Acid Schiff, for staining mucins. (**H, I**) Quantification of immunohistochemical staining of treated mice pERK (**H**) and pAKT (**I**) as described in 'Materials and methods'. Statistical analysis was measured with an ordinary one-way ANOVA. *p<0.05, **p<0.01, ns = not significant. N = 4–5 mice.

The online version of this article includes the following figure supplement(s) for figure 4:

**Figure supplement 1.** MicroCT, body weight, and pathological analysis of treated KL mice.

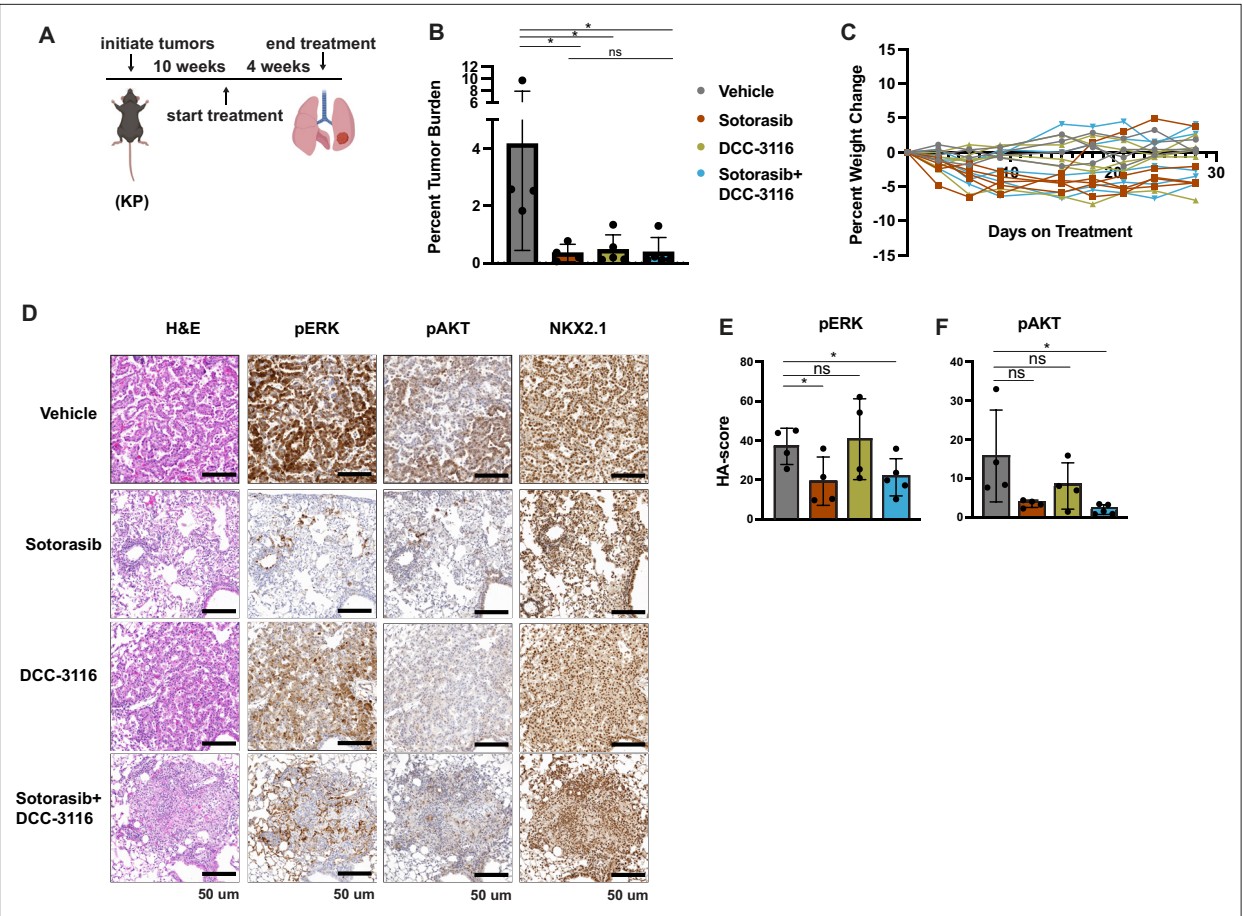

**Figure 5.** Inhibition of KRAS[G12C] and ULK1/2 reduces tumor burden in a KP genetically engineered mouse (GEM) model. (**A**) Schematic of the treatment of KP GEM models. Panel (**A**) was created with BioRender.com and published using a CC BY-NC-ND license with permission. Mice were administered vehicle control or 30 mg/kg sotorasib once daily via oral gavage. DCC-3116 was administered in drug-formulated chow. N = 4–5 mice. (**B**) Quantification of tumor burden of mice after 4 weeks of treatment. Statistical analysis was performed using an ordinary one-way ANOVA. *p<0.05, ns = not significant. N = 4–5 mice. (**C**) Percent change in the body weight of mice on treatment over 4 weeks. Each line depicts an individual mouse. (**D**) Representative images of histological analysis of lung lobes from KP mice 4 weeks after treatment. (**E, F**) Quantification of immunohistochemical staining of treated mice pERK1/2 (**E**) and pAKT1-3 (**F**) as described in 'Materials and methods'. Statistical analysis was performed using an ordinary one-way ANOVA. *p<0.05, **p<0.01. ns = not significant. N = 4–5 mice.

The online version of this article includes the following figure supplement(s) for figure 5:

**Figure supplement 1.** MicroCT images of KP mice on treatment.

---

weight loss in mice in any of the three treatment groups (**Figure 5C**). Immunohistochemical analysis of lung sections from mice again revealed decreased pERK1/2 in mice treated with sotorasib either alone or in combination with DCC-3116, although pERK1/2 was detected in some areas of the lung (**Figure 5D and E**). Vehicle-treated KRAS[G12C]/TP53[R172H]-driven lung tumors had detectable pAKT, which was decreased by all three treatments (**Figure 5D and F**). Finally, all tumors displayed expression of NKX2.1 that was not substantially altered by drug treatment (**Figure 5D**).

## KL lung cancer-derived cells that acquire resistance to sotorasib increase RAS and pERK1/2 expression and do not increase autophagy after sotorasib treatment

To complement our in vivo work with GEM models and better understand how acquired sotorasib resistance influences autophagy and treatment sensitivity, we generated KRAS[G12C]/LKB1[Null]-lung cancer-derived cell lines from suitably manipulated, drug-naive KL mice. As expected, parental

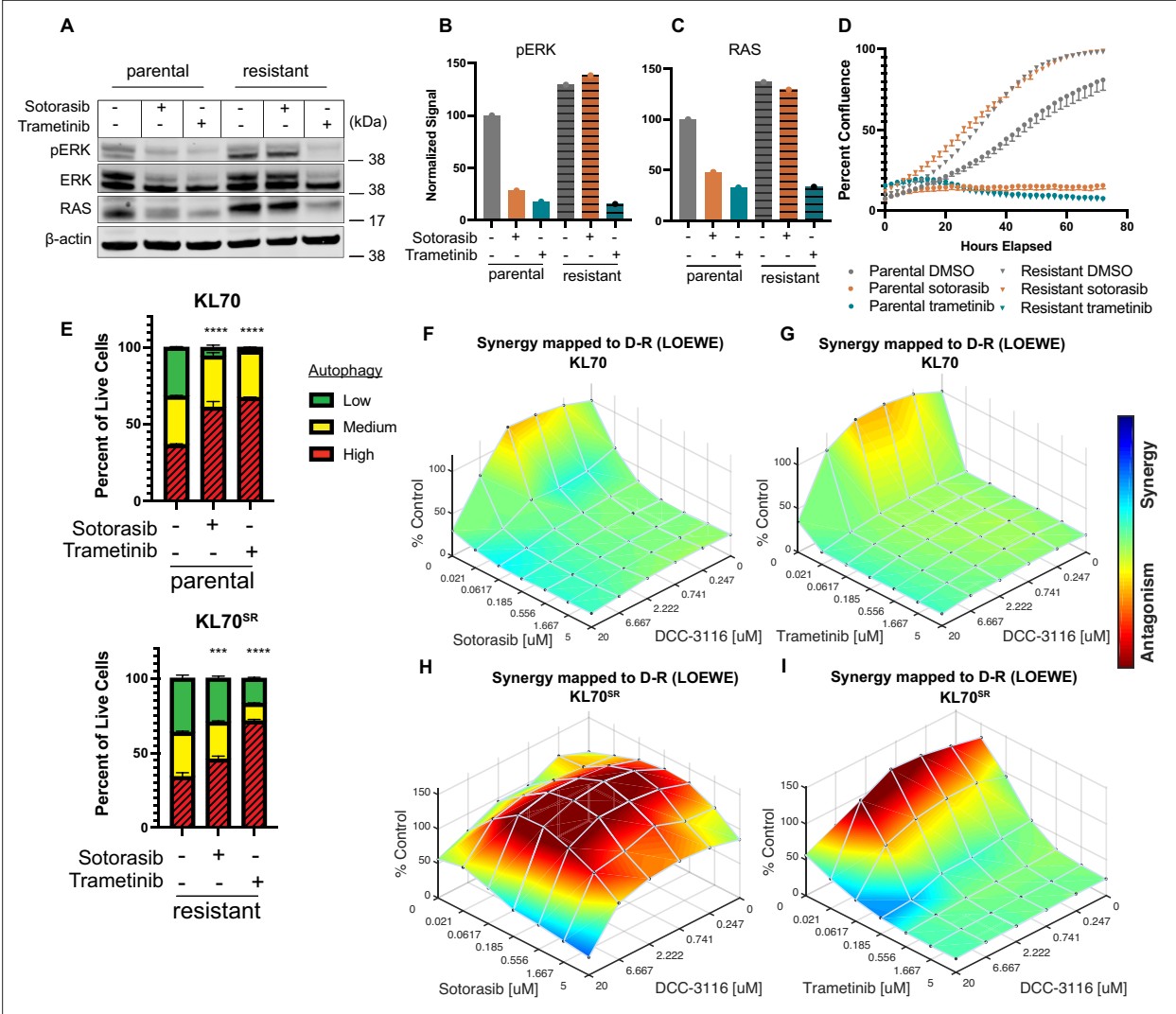

**Figure 6.** KL lung-cancer-derived cells that acquire resistance to sotorasib increase RAS and pERK1/2 expression and do not increase autophagy after sotorasib treatment. (**A**) Immunoblot analysis of KL.70 and KL.70[R] cells treated with 100 nM sotorasib or 100 nM trametinib after 48 hr of treatment. (**B, C**) Quantification of signal from A normalized to b-actin. (**D**) Live-cell imaging of percent confluence of KL.70 cells over time treated with DMSO, 100 nm sotorasib of 100 nM trametinib. N = 3 biological replicates. (**E**) Autophagy measurement with fluorescent autophagy reporter (FAR) in cells assessed by mCherry-eGFP-LC3 reporter after 48 hr of 100 nM sotorasib or 100 nM trametinib treatment. Red = high autophagy, yellow = medium autophagy, green = low autophagy. Statistical significance was determined by comparing autophagy levels to DMSO control, and an ordinary one-way ANOVA with Dunnett's multiple comparisons was used. Ns = not significant, *p<0.05, **p<0.01, ***p<0.001, ****p<0.0001. N = 9 biological replicates. (**F, G**) In vitro synergy assay of KL70 cells treated with indicated doses of sotorasib and/or DCC-3116 using the Loewe method after 72 hr of treatment. N = 3 biological replicates. (**H, I**) In vitro synergy assay of KL70[SR] cells treated with indicated doses of sotorasib and/or DCC-3116 using the Loewe method after 72 hr of treatment. N = 3 biological replicates.

The online version of this article includes the following source data for figure 6:

**Source data 1.** Uncropped and labeled gels for *Figure 6*.

**Source data 2.** Raw unedited gels for *Figure 6*.

KL70 cells are sensitive to the antiproliferative effects of both sotorasib and the MEK1/2 inhibitor, trametinib (*Figure 6A–D*). In addition, pERK1/2 also is inhibited by both sotorasib and trametinib (*Figure 6A*). Next, we generated a sotorasib resistant (SR) KRAS[G12C]/LKB1[Null]-driven lung cancer cell line by culturing parental KL70 cells over 12 weeks in gradually increasing concentrations of sotorasib until a resistant population emerged (KL70[SR] cells). KL70[SR] cells were entirely resistant to the antiproliferative effects of sotorasib but remained sensitive to trametinib (*Figure 6D*). Consistent with this, the level of pERK1/2 was sotorasib resistant but trametinib sensitive (*Figure 6A*). Indeed, KL70[SR] cells had

a higher baseline level of both RAS and pERK1/2 expression compared to parental KL70 cells, which might explain their more rapid baseline proliferation (*Figure 6D*).

We next tested the differences in baseline and drug-induced autophagy in KL70 and KL70[SR] cells using the FAR. As expected based on analysis of human KRAS[G12C]-driven lung cancer cells, KL70 cells increased autophagy in response to either sotorasib or trametinib treatment, a further demonstration that LKB1 is not required for increased autophagy in response to inhibition of KRAS[G12C]>RAF>MEK>ERK inhibition (*Figure 6E*). KL70[SR] cells treated with sotorasib displayed little induction of autophagy (*Figure 6E*). However, trametinib elicited a robust induction of autophagy in KL70[SR] cells (*Figure 6E*), further suggesting that changes in autophagy levels are correlated to pERK1/2 levels in the cell. Because sotorasib did not induce autophagy in the KL70[SR] cells, it was not clear whether the addition of an autophagy inhibitor would decrease cell viability in the SR cells. Cell viability was measured after 72 hr of drug treatment in the KL70 and KL70[SR] cell lines (*Figure 6F–I*). The combination of sotorasib and DCC-3116 did not lead to decreases in cell viability in the KL70[SR] cells (*Figure 6H*). We next tested whether there were any synergistic decreases in cell viability with the combination of trametinib and DCC-3116 because there was a robust increase in autophagy levels with trametinib treatment in the KL70[SR] cells. KL70[SR] cells are exquisitely sensitive to trametinib at very low concentrations, and therefore, little synergy was observed with the tested concentrations (*Figure 6I*).

## Discussion

Cancer cells exhibit increased autophagy in response to a variety of cellular stresses including nutrient deprivation or the application of pathway-targeted therapy, thereby allowing cancer cells to recycle a wide variety of macromolecules and organelles to promote cell viability. (*Kinsey et al., 2019*; *Bhatt et al., 2023*) Moreover, GEM models of lung or pancreatic cancer indicate that cancer cells employ autophagy – both cancer cell autonomous and from the mouse as a whole – at several stages of cancer progression (*Rao et al., 2014*; *Li et al., 2021a*). Recently, we and others have demonstrated that many RAS-driven cancer cells display increased autophagy in response to blockade of RAS>RAF>MEK>ERK signaling (*Kinsey et al., 2019*; *Bryant et al., 2019*; *Bhatt et al., 2023*). Moreover, in our work, combined inhibition of MEK1/2 (with trametinib) plus inhibition of lysosome function (with CQ or HCQ) displayed synergistic antiproliferative activity in vitro, superior tumor control (compared to the single agents or standard of care) in cell line- or patient-derived xenografts (CDX or PDX) models of pancreatic cancer and, in one patient with advanced pancreatic cancer, there was evidence of antitumor activity of the combination of trametinib plus HCQ (*Kinsey et al., 2019*). These data (and others) have led to a number of clinical trials, some of which are ongoing (clinicaltrials.gov: NCT04214418, NCT04132505, NCT04386057, NCT04145297). Here, we expand these observations using preclinical models of KRAS[G12C]-driven lung cancer. The management of this disease has been transformed by FDA approval of adagrasib and sotorasib, covalent inhibitors of the KRAS[G12C] oncoprotein (*Canon et al., 2019*; *Fell, 2020*; *Hallin et al., 2020*). However, as is generally true for single-agent pathway-targeted cancer therapy, the emergence of drug-resistant disease can be rapid and pleiotropic (*Ashrafi et al., 2022*). Our data suggest that combined inhibition of KRAS[G12C] plus autophagy may have superior activity against KRAS[G12C]-driven lung cancer. These data are in accordance with a recent study in which HCQ-mediated inhibition of lysosome function sensitized a KRAS[G12D]/LKB1[Null]-driven GEM model of lung cancer to MEK1/2 inhibition (*Bhatt et al., 2023*). An interesting aspect of these results is the apparent dispensability of LKB1 for the induction of autophagy in response to inhibition of KRAS signaling. Our work and that of others had concluded that LKB1 was important for trametinib-induced autophagy in pancreatic cancer cells, but this new data argues that LKB1 is not an obligate requirement for autophagy induction (*Kinsey et al., 2019*). Co-alterations in KRAS plus LKB1 are frequent in lung cancer, and patients with these alterations are reported to display resistance to current therapeutic options such as chemotherapy and/or immunotherapy (*Sanchez-Cespedes, 2011*; *Sanchez-Cespedes et al., 2002*; *Caiola et al., 2018*). In addition, and perhaps surprisingly, we noted that DCC-3116 had substantial single-agent activity against KRAS[G12C]-driven lung cancers in GEM models. Importantly, previous studies have demonstrated that combined inhibition of MEK1/2 plus autophagy does not lead to synergistic or cooperative decreases in cell proliferation or tumor growth in a GEM model of KRAS[G12D]/TP53[Null]-driven lung cancer in which LKB1 expression is retained (*Bhatt et al., 2023*). Here, we show that combined inhibition of KRAS[G12C] plus ULK1/2 in Calu-1 cells (KRAS[G12C]/LKB1[WT]/TP53[null]) decreased cell proliferation and tumor growth and that this combination

was more effective than either single agent. Our findings demonstrate that inhibition of both KRAS[G12C] plus ULK1/2 kinases is synergistic in KRAS[G12C] mutant lung cancer cells with wild-type LKB1 expression but loss of TP53 expression. The difference between these findings could be due to the difference in KRAS missense mutation driving lung tumorigenesis or differences between inhibiting autophagy with a 4-aminoquiniolone such HCQ and DCC-3116-mediated inhibition of ULK1/2. Indeed, since HCQ inhibits lysosome function, it might inhibit cellular recycling of substrates acquired by micropinocytosis, whereas ULK1/2 inhibition should be specific for autophagic recycling of intracellular substrates. Hence, this work supports the hypothesis that patients with KRAS[G12C]-driven lung cancers may benefit from treatment with the combination of KRAS[G12C] plus ULK1/2 inhibitors. To that end, there is an ongoing clinical trial enrolling patients with KRAS[G12C]-driven cancers to test the combination of sotorasib plus DCC-3116 (NCT04892017).

Autophagy is reported to promote lung tumor progression by supporting tumor energy production (*Bhatt et al., 2019*). Others have demonstrated that silencing of autophagy-related 7 (ATG7) expression, a protein essential for autophagy, in KRAS[G12D]/LKB1[null]-driven tumors at the time of tumor initiation substantially diminishes lung tumor burden (*Bhatt et al., 2019*). Our study suggests that pharmacological inhibition of ULK1/2 kinase activity starting at tumor initiation also significantly reduces growth of KRAS[G12C]/LKB1[null] lung tumors. Together, these data suggest that both ATG7 and ULK1/2 kinases are necessary for KRAS-driven lung tumorigenesis even when LKB1 expression is silenced. Intriguingly, treatment of mice bearing established KRAS[G12C]-driven lung tumors with single-agent DCC-3116 decreased tumor burden, further demonstrating that ULK-mediated autophagy may be important for maintenance of KRAS[G12C]-driven lung tumors. Although we and others have demonstrated that autophagy inhibition as a monotherapy modestly reduces tumor burden in some preclinical models (*Bhatt et al., 2023*; *Bhatt et al., 2019*), this has not been recapitulated in human cell line-derived models or in clinical trials with HCQ. This could be attributed to the relatively low potency of autophagy inhibition with HCQ, the low mutational burden of tumors arising in GEM models compared to human cancers or the use of nonspecific autophagy inhibitors (*Wolpin et al., 2014*; *Mancias and Kimmelman, 2011*; *Amaravadi et al., 2019*). Indeed, we did observe that DCC-3116 treatment modestly inhibited the growth of one human KRAS[G12C] mutant lung cancer cell line, NCI-H358, both in vitro and in vivo, suggesting that ULK1/2 inhibition as a monotherapy may indeed have significant single-agent activity in certain settings.

In addition to the assessment of preclinical therapeutic responses in GEM models, this study also addressed the histopathology of KRAS[G12C]-driven lung tumors, either in combination with LKB1 silencing or expression of dominant-negative TP53[R172H], in GEM models. In both the KRAS[G12C]/LKB1[Null] and KRAS[G12C]/TP53[R172H] GEM models of lung tumorigenesis, we noted the outgrowth of AAH and adenocarcinomas as was reported for KRAS[G12D]-driven lung tumors some years ago. Interestingly, KRAS[G12C]/LKB1[Null]-driven lung tumors also displayed mixed ASC and mucinous adenocarcinomas that were not observed in the KRAS[G12C]/TP53[R172H]-driven model. Analysis of the mucinous adenocarcinomas indicated that some tumor cells expressed the gastric lineage marker HNF4α, an observation originally reported in KRAS[G12D]/NKX2.1[Null]-driven lung tumors *Snyder et al., 2013*; *Camolotto et al., 2018*. In this (and related) GEM model(s), silencing of NKX2.1, a master transcriptional regulator of both lung and thyroid development, promoted the transition of lung tumor cells to a mucinous gastric cell phenotype through the action of HNF4α in partnership with the FOXA1/2 transcription factors (*Camolotto et al., 2018*). Interestingly, HNF4α-expressing KRAS[G12C]/LKB1[Null]-driven lung tumor cells retained expression of NKX2.1, suggesting that NKX2.1 silencing is not required for the emergence of mucinous adenocarcinomas when LKB1 expression is silenced. Whether this reflects the ability of LKB1 signaling to (1) regulate cancer metabolism through effects on triosephosphate isomerase or altered lactate production and secretion, (2) regulate transcription effects on histone acetylation via elevated CRTC2-CREB signaling downstream of the salt-inducible kinases or effects on HDAC1/3 or some other signaling circuit remains to be determined (*Stein et al., 2023*; *Eichner et al., 2023*; *Hollstein et al., 2019*; *Shackelford and Shaw, 2009*; *Qian et al., 2023*; *Compton et al., 2023*).

Finally, in both GEM models of KRAS[G12C]-driven lung tumorigenesis, we observed inhibition of both AAH and adenocarcinoma with either single-agent treatment or the combination of sotorasib plus DCC-3116. By contrast, in the GEM model of KRAS[G12C]/LKB1[Null]-driven lung tumorigenesis, we noted an enrichment of the mixed adenosquamous and mucinous carcinomas after cessation of therapy. This suggests that this histopathological class of KRAS[G12C]/LKB1[Null]-driven lung tumors may be relatively

resistant to agents such as sotorasib that covalently inhibit the inactive, GDP-bound state of KRAS[G12C]. This could reflect (1) the differences in expression of RGS GTPase-activating proteins (GAPs) that promote GTP hydrolysis by KRAS[G12C]; a cell state transition that is less dependent on the KRAS[G12C] oncoprotein as has been described for resistance of EGFR-driven lung cancers to pathway-targeted inhibition of EGFR tyrosine kinase activity; or (2) elevated expression of downstream factors such as c-MYC that we have recently shown can confer resistance to the combination of trametinib plus HCQ in models of KRAS-driven pancreatic cancer (*Silvis et al., 2023*; *Li et al., 2021b*; *Sequist et al., 2011*).

In closing, our data support the testing of ULK1/2 inhibitors in patients with KRAS-driven lung cancer, either as single agents or in combination with direct inhibitors of KRAS oncoproteins, or of KRAS-regulated signaling pathways downstream of KRAS such as the RAF>MEK>ERK MAP kinase pathway. It will be interesting to interrogate specimens from patients on such trials prior to and after the likely eventual emergence of drug resistance to determine the extent to which potential resistance mechanisms glimpsed in preclinical models are observed in clinical trials. Such observations may pave the way for more efficacious regimens of therapy that maximize the depth and durability of desirable antitumor effects while minimizing toxicity.

## Materials and methods
### Cell culture
Cell lines were routinely tested for mycoplasma contamination. All human lung cancer cell lines were obtained from the ATCC and cultured in RPMI 1640 (Gibco 11875-093) supplemented with 10% (v/v) fetal bovine serum (FBS) (Gibco 10438-026) and 1% penicillin plus streptomycin (P/S) (Gibco 15140-122). Mouse tumor-derived cell lines were derived as previously described (PMID:19282848). Briefly, lungs from tumor-bearing mice were homogenized in a solution of digestive enzymes and plated in dishes. Cells were harshly split to allow for the outgrowth of tumor cells. The specific genetic abnormalities in each cell line were confirmed by analysis of genomic DNA or by immunoblotting. All mouse tumor-derived cell lines were cultured in Dulbecco's Modified Eagle Medium F12 (DMEM/F12) (Gibco 11330-032) with 10% (v/v) FBS and 1% P/S. KL.70 cells were derived from a 24-week-old female mouse.

### Fluorescent autophagy reporter
A vector encoding the chimeric FAR protein comprising mCherry:EGFP:LC3 was introduced into human and mouse cell lines as previously described with virus-infected cells selected using puromycin and FAR-expressing cells selected by flow cytometry (*Kinsey et al., 2019*; *Kimura et al., 2007*). To assess autophagy, FAR-expressing cells were seeded at equal densities in 6-well plates. After 24 hr, media was replaced with media containing the various drug treatments for 48 hr. Cells were harvested and stained with 1 mM SYTOX Blue (Invitrogen S34857) for the exclusion of dead cells. Flow cytometry (Beckman Coulter CytoFLEX) was performed to assess mCherry and EGFP intensity and quantify the mCherry:EGFP ratio normalized to dimethyl sulfoxide (DMSO) control. Cells engineered to express the FAR are analyzed by flow cytometry in which we defined autophagy status by gating viable (based Sytox Blue staining), DMSO-treated control cells into three bins based on the ratio of EGFP:mCherry fluorescence. We gate all live cells into the 33% highest EGFP-positive cells (autophagy low) and the 33% highest mCherry-positive cells (autophagy high), and therefore, the proportion in the middle is also approximately 33% and considered the medium autophagy status. Again, these gates are based entirely on the DMSO-treated control cells, and all other treatments within the experiment are compared to settings on these gates. For statistical analysis, an ordinary one-way ANOVA and Dunnett's multiple comparisons test were performed.

### Lentiviral transduction
HEK293T cells (Key resources table) were seeded 24 hr prior to transfection in DMEM/F12 (10% FBS 1% P/S). DNA for lentivirus generation was introduced into these cells using a Lipofectamine 3000 kit (Invitrogen L3000015). All virus-containing supernatants were filtered through 0.45 mm filters before use. To increase the efficiency of infection, 10 mg/mL of Polybrene (MilliporeSigma TR-1003-G) was supplemented in the virus-containing media when added to cells. Cells were selected for successful infection through antibiotic selection with the corresponding antibiotics (puromycin or blasticidin) or

flow cytometry for fluorescent markers. All plasmids used are described in Key resources table and are commercially available or previously published except the pLV-ULK1$^{K46N}$ plasmid.

## ELISA for pS318-ATG13

pS318-ATG13 antibody (Rockland Immunochemicals, 600-401C49) was biotinylated using a Biotin Labeling Kit-NH$_2$ (Dojindo, LK03). ELISA plates (Corning, 9018) were coated in anti-ATG13 antibody (Key resources table) overnight at 4°. Cells were plated in equal densities in a 96-well plate. After 24 hr, cells were treated with various drug treatments for 16 hr, washed with 1× phosphate-buffered saline, and then lysed in M-Per lysis buffer (Thermo Fisher, 78501) supplemented with 1× ethylene-diaminetetraacetic acid (Thermo Fisher, 1861274), 1× Sigma Phosphatase Inhibitor (Sigma, 5726), 2× Halt Phosphatase and Protease inhibitor Cocktail (Thermo Fisher, 78446) for 15 min on an orbital shaker at 4°C. Cell extracts were then centrifuged for 10 min at 4000 rpm in a 4°C refrigerated centrifuge. ELISA plates were washed twice with 1× ELISA wash buffer (BioLegend, 421601), blocked for 1 hr at room temperature (RT) with 1× ELISA dilutant buffer (BioLegend Cat# 421203), then washed twice with 1× wash buffer. Cell extracts were diluted in wash buffer, incubated in the ELISA plate at RT for 2 hr, and then washed twice with 1× wash buffer. The biotinylated pS318-ATG13 antibody was diluted in 1× dilutant buffer and added to each well. After 1 hr at RT, wells were washed twice with 1× well wash buffer. Streptavidin-Poly-HRP antibody (Key resources table) was diluted in 1× dilutant and added to each well for 1 hr at RT. Wells were washed twice with 1× well wash buffer and tetramethylbenzidine substrate (BioLegend, 421101) was added to each well for 15 min at RT. After 15 min, ELISA stop solution (Invitrogen, SS04) was added to each well. Background correction was performed by including control wells that received every reagent except for protein lysate and subtracting that value from all samples. Plates were immediately read on a Synergy HTX (BioTek) plate reader at 450 nm and 540 nm. Statistical analysis was performed using a one-way ANOVA with Tukey's multiple comparisons.

## Immunoblotting

Cell extracts were harvested as previously described (*Truong et al., 2020*). Following clearing by centrifugation, protein lysates were quantified using a bicinchoninic acid assay (Thermo Scientific 23250). 20-50 mg samples of cell extract were fractionated by SDS-PAGE and then western blotted onto nitrocellulose membranes. Membranes were probed with primary antibodies (Key resources table) overnight at 4°C. Secondary antibodies (Key resources table) were diluted and incubated at RT for 1 hr. Membranes were imaged and analyzed on an Odyssey CLx Infrared Imaging System (LI-COR) and analyzed using Image Studio Software (LI-COR).

## In vitro synergy assays

Cells were seeded in triplicate at 5000–7000 cells/well into black-walled clear bottom 96-well plates (Costar 3603), cultured overnight in complete media, and then treated with various drug treatments at the indicated concentrations for 72 hr. At the endpoint, ATP was quantified using CellTiter-Glo luminescent cell viability assay (Promega, G7570) as per the manufacturer's instructions. Luminescence was measured using a BioTek Synergy HTX plate reader and data were normalized to untreated controls. Synergy was determined using ComBenefit software (HSA, Bliss, Loewe models) (*Di Veroli et al., 2016*).

## Live-cell imaging

Cells were imaged using an IncuCyte Zoom Live Cell Imager (Sartorius) over time and analyzed using the IncuCyte Analysis Software (Sartorius). Cells were seeded at equal densities in 6-well plates and treated with various inhibitors either as single agents or in combination and imaged every 2 hr. Media was replaced after 48 hr for each well. Percent confluence and standard error were calculated at each time point. Statistical analysis was performed using a one-way ANOVA with Tukey's multiple comparisons.

### Generation of pLV-ULK$^{K46N}$ expression vector

The pLV-ULK1$^{K46N}$ plasmid was designed and constructed using VectorBuilder. 3× HA-tagged human ULK1$^{K46N}$ was cloned into a pLV lentivirus backbone vector containing a tetracycline-inducible promoter including a Tet response element and puromycin resistance cassette synthesized.

### Animal work

All animal work was approved by the Institutional Care and Use Committees at the University of Utah (protocol # 21-1005). All work containing biohazardous agents was approved by the University of Utah Biosafety Committee. Mice were fed ad libitum, housed in micro isolator cages, and monitored daily in a temperature-controlled environment with a 12-hr light/dark cycle. *Kras*$^{LSL-G12C}$ and Lkb1$^{fl/fl}$ mice were purchased from The Jackson Laboratory (Key resources table; *Zafra et al., 2020*; *Nakada et al., 2010*). All mice were on a mixed genetic background. Both female and male mice were used for all experiments. Experimental adult mice were initiated between 6 and 8 weeks of age through nasal inhalation of adenovirus expressing CMV-Cre recombinase (Key resources table) using $5 \times 10^7$ pfu per mouse as previously described (*Fasbender et al., 1998*). Lung tumorigenesis was monitored in live mice with micro-computed tomography (microCT) scans (Perkin Elmer Quantum Gx) with a Cu 0.06 + AI 0.5 X-ray filter, 72 mm acquisition, and 45 mm reconstitution for a 2 min standard scan. Mice were scanned up to nine times (<1 gray of total radiation). Mice on treatment were weighed 2–3× per week and euthanized if unacceptable weight loss (>20%) was observed. Upon euthanasia, lungs were dissected and fixed using standard protocols (*Limjunyawong et al., 2015*). For xenograft models, Nod.Cg-Prkdcscid/J mice were purchased from The Jackson Laboratory (Key resources table) or generated in-house. Cell lines were injected subcutaneously into the flank of adult mice at $1–5 \times 10^6$ cells per injection in a 1:1 mixture of serum-free Opti-MEM (Gibco, 31985-070) and Matrigel (Corning, 356231). Once tumors were palpable, they were measured using digital calipers. Once tumors reached ~250 mm$^3$, mice were randomized to treatment arms. Mice were treated, in a nonblinded manner, for 28 days or until maximum tumor volume was reached (~1 cm$^3$). Tumor volume was calculated as equal to (tumor length) × ((tumor width$^2$)/2). Statistical analysis on xenograft assays was performed using an ordinary one-way ANOVA and unpaired Student's *t*-test. Corn oil vehicle (Sigma C8261) and sotorasib treatments were administered once daily via oral gavage. DCC-3116 was formulated in chow, and mice were fed ad libitum. Schematics illustrating treatment strategies were created using BioRender.

### Survival of mice on treatment

KL mice were randomized to treatment arms at 10 weeks post tumor initiation (p.i.) and treated every day or until euthanasia end point criteria were reached or after a predetermined 56 days of drug treatment. Euthanasia criteria were defined per IACUC protocol to monitor pain and distress symptoms in mice (*Ullman-Culleré and Foltz, 1999*). Statistical analysis of survival was performed using the log-rank (Mantel–Cox) test.

### Tissue hematoxylin and eosin, immunohistochemistry staining

Lungs were harvested and inflated by perfusion through the trachea with 5 mL of 10% (v/v) buffered formalin (Epredia 5725), and then fixed overnight with gentle rocking at RT following standard protocols (*Limjunyawong et al., 2015*). Fixed lungs were embedded in paraffin and then 5 µm sections were prepared. Harris hematoxylin acidified (Epredia 6765003) and Eosin Y (Epredia 71211) stains were performed on all tissue sections. For immunohistochemistry (IHC), sections were first deparaffinized using standard protocols, and antigen retrieval was performed in a pressure cooker using 1× Citrate Buffer pH 6 (Novus Biologicals, NB90062075). IHC was performed using the ImmPRESS Polymer Reagents (Vector Laboratories). Primary antibodies (Key resources table) were diluted and incubated at RT for 1 hr and then counterstained with hematoxylin. Sections were dehydrated using standard protocols and mounted with ClearVue Mountant (Epredia 4212). Alcian Blue – Periodic Acid Schiff (PAS) staining was performed using standard protocols. Briefly, slides were deparaffinized, rehydrated, and incubated in Alcian Blue (Leica 38016SS3A). Slides were washed and incubated in periodic acid (Leica 38016SS4A) followed by the Schiff Reagent (Fisher SS32-500).

### Tissue imaging and quantification

All tissue imaging was performed using a 3D-Histech Midi Panoramic Scanner (Epredia) at ×20 magnification. Tumor burden was quantified using CaseViewer software (Epredia). Total lung area and tumor

area were digitally annotated using a closed polygon annotation. Total tumor and lung areas were summed, and the percent tumor burden was calculated. Immunohistochemical quantification was performed using 3D-Histech Quant Center software (Epredia) as previously described (*Silvis et al., 2023*). Statistical analysis was performed using an ordinary one-way ANOVA.

## Inhibitor treatments

All pathway-targeted inhibitors (Key resources table) were obtained from reputable sources and resuspended in the appropriate solvent (DMSO or water) for in vitro testing. The biochemical activity of all compounds was tested in vitro before being used in vivo. Compounds for in vivo dosing were resuspended in 10% (v/v) DMSO and 90% (v/v) corn oil (Key resources table) or formulated in mouse chow. The DCC-3116 formulated mouse chow (Research Diets) was formulated with an OpenStandard Diet with 15% kcal% fat and 360 mg DCC-3116 CL/kg diet.

## Materials availability

Any newly created materials, such as the KL.70 cells, will be made available to any investigator by reaching out to the corresponding author.

## Acknowledgements

We want to acknowledge the members of the McMahon and Kinsey labs for their valuable support, guidance, and sharing of reagents. PCG was supported by a JEDI award from the Life Sciences Editors Foundation, and we graciously thank Li-Kuo Su for editing suggestions and advice. The content is solely the responsibility of the authors and does not necessarily represent the official views of the National Institutes of Health. This work was supported by the University of Utah Flow Cytometry Facility and the National Cancer Institute (NCI) through award number P30 CA042014. PCG would like to acknowledge funding provided by NCI under award number F31 CA261116. MMcM would like to acknowledge funding provided by NCI under award numbers R01 CA131261, R01 CA176839, and P30 CA042014, the Five for the Fight, the Huntsman Cancer Foundation, and the Huntsman Lung Cancer Disease Center. CGK would like to acknowledge funding provided by NCI award number K08 CA230151, the Huntsman Cancer Foundation, and the V Foundation Clinical Scholar Award. ELS would like to acknowledge funding provided by the NCI under award numbers R01 CA212415, R01 CA240317, and R01 CA237404. GL would like to acknowledge funding provided by the NCI under award number R01 CA82577.

## Additional information

### Competing interests

Madhumita Bogdan, Bryan D Smith, Daniel L Flynn: stockholder of Deciphera Pharmaceuticals. Conan G Kinsey, Martin McMahon: Research described here was supported through a Sponsored Research Agreement between the University of Utah and Deciphera Pharmaceuticals award to MM and CK. The other authors declare that no competing interests exist.

### Funding

| Funder | Grant reference number | Author |
|---|---|---|
| National Cancer Institute | F31CA261116 | Phaedra C Ghazi |
| National Cancer Institute | R01CA131261 | Martin McMahon |
| National Cancer Institute | R01CA176839 | Martin McMahon |
| National Cancer Institute | P30CA042014 | Martin McMahon |
| Huntsman Cancer Foundation | | Martin McMahon Conan G Kinsey |
| National Cancer Institute | K08CA230151 | Conan G Kinsey |

| Funder | Grant reference number | Author |
|---|---|---|
| Huntsman Cancer Institute, University of Utah | | Martin McMahon |
| V Foundation for Cancer Research | | Conan G Kinsey |
| National Cancer Institute | R01CA212415 | Eric L Snyder |
| National Cancer Institute | R01CA240317 | Eric L Snyder |
| National Cancer Institute | R01CA237404 | Eric L Snyder |
| National Cancer Institute | R01CA82577 | Guillermina Lozano |

The funders had no role in study design, data collection and interpretation, or the decision to submit the work for publication.

## Author contributions

Phaedra C Ghazi, Conceptualization, Data curation, Formal analysis, Funding acquisition, Validation, Investigation, Visualization, Methodology, Writing - original draft, Writing - review and editing; Kayla T O'Toole, Sanjana Srinivas Boggaram, Michael T Scherzer, Data curation, Writing - review and editing; Mark R Silvis, Data curation, Methodology, Writing - review and editing; Yun Zhang, Guillermina Lozano, Resources; Madhumita Bogdan, Daniel L Flynn, Conceptualization, Resources, Writing - review and editing; Bryan D Smith, Conan G Kinsey, Conceptualization, Writing - review and editing; Eric L Snyder, Formal analysis, Writing - review and editing; Martin McMahon, Conceptualization, Supervision, Funding acquisition, Investigation, Visualization, Methodology, Writing - original draft, Writing - review and editing

## Author ORCIDs

Phaedra C Ghazi  https://orcid.org/0000-0002-0884-9442
Guillermina Lozano  https://orcid.org/0000-0001-8985-4886
Eric L Snyder  https://orcid.org/0000-0003-3591-3195
Conan G Kinsey  https://orcid.org/0000-0001-5614-8627
Martin McMahon  https://orcid.org/0000-0003-2812-1042

## Ethics

All animal work was approved by the Institutional Care and Use Committees at the University of Utah (protocol #: 21-1005) and handled as outlined in the protocol. Every effort was made to minimize suffering.

Reviewer #2 (Public Review): https://doi.org/10.7554/eLife.96992.3.sa1
Author response https://doi.org/10.7554/eLife.96992.3.sa2

# Additional files

## Supplementary files

• MDAR checklist

## Data availability

All data generated or analyzed during this study are included in the manuscript and supporting files; source data files have been provided for Figure 1, Figure 2, and Figure 1-figure supplement 2.

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
