## [Editor Report · eLife assessment]

This is a mechanistic study showing the effect of combining inhibition of autophagy (through ULK1/2) and KRAS (using sotorasib) on KRAS mutant NSCLC making the study **valuable** to cancer biologists and more broadly in a clinical setting. The evidence generated by GEM mouse models and cell lines is **solid** but could be further strengthened by increasing the mouse cohort size. This study holds translational relevance beyond NSCLC to other indications that carry KRAS mutations.

---

## [Referee Report · Reviewer #2 (Public Review)]

Summary:

In this manuscript, Ghazi et reported that inhibition of KRASG12C signaling increases autophagy in KRASG12C expressing lung cancer cells. Moreover, the combination of DCC 3116, a selective ULK1/2 inhibitor, plus sotorasib displays cooperative/synergistic suppression of human KRASG12C driven lung cancer cell proliferation in vitro and tumor growth in vivo. Additionally, in genetically engineered mouse models of KRASG12C driven NSCLC, inhibition of either KRASG12C or ULK1/2 decreases tumor burden and increases mouse survival. Additionally, this study found that LKB1 deficiency diminishes the sensitivity of KRASG12C/LKB1Null-driven lung cancer to the combination treatment, perhaps through the emergence of mixed adeno/squamous cell carcinomas and mucinous adenocarcinomas.

Strengths:

Both human cancer cells and mouse models were employed in this study to illustrate that inhibiting ULK1/2 could enhance the responsiveness of KRASG12C lung cancer to sotorasib. This research holds translational importance.

Weaknesses:

The revised manuscript has addressed most of my previous concerns. However, I still have one issue: the sample size (n) for the GEMM study in Figures 4E and 4F is too small, despite the authors' explanation. The data do not support the conclusion due to the lack of significant difference in tumor burden. Additionally, the significance labels in Figure 4E are not clearly explained.

---

## [Author Response]

The following is the authors’ response to the original reviews.

**Public Reviews:**

**Reviewer #1 (Public Review):**
Summary:Given that KRAS inhibition approaches are a relatively new innovation and that resistance is now being observed to such therapies in patients with NSCLC, investigation of combination therapies is valuable. The manuscript furthers our understanding of combination therapy for KRAS mutant non-small cell lung cancer by providing evidence that combined inhibition of ULK1/2 (and therefore autophagy) and KRAS can inhibit KRAS-mutant lung cancer growth. The manuscript will be of interest to the lung cancer community but also to researchers in other cancer types where KRAS inhibition is relevant.Strengths:The manuscript combines cell line, cell line-derived xenograft, and genetically-engineered mouse model data to provide solid evidence for the proposed combination therapy. The manuscript is well written, and experiments are broadly well performed and presented.

We thank Reviewer #1 (R1) for the generally favorable review of our manuscript, and also for the more detailed critique that identifies potential weaknesses in the research, which we address on a point-by-point basis below.

Weaknesses:With 3-4 mice per group in many experiments, experimental power is a concern and some comparisons (e.g. mono vs combination therapy) seem to be underpowered to detect a difference. Both male and female mice are used in experiments which may increase variability.

We thank R1 for pointing out concerns regarding statistical power in our various mouse models of NSCLC experiments, and agree that more mice per group would certainly increase statistical power. However, there are certain logistical considerations that impact the generation of cohorts of experimental *KrasLSL-G12C* mice. Because mice homozygous for the *KrasLSL-G12C* allele display embryonic lethality, we are required to generate experimental mice by crossing heterozygous male and female *KrasLSL-G12C* mice. Although 66% of the progeny of such crosses are predicted to be *KrasLSL-G12C/+*, experience tells us that we only obtain ~40-50% heterozygous *KrasLSL-G12C/+* mice with litter sizes around 6-8 mice from such crosses. Therefore, there are usually only about 4 heterozygous *KrasLSL-G12C* mice per litter, which presents a substantial challenge in generating larger cohorts of age-matched mice suitable for experiments, especially under conditions where we wish to euthanize mice at multiple time points for analysis. For the GEM model experiments, Figure 3B is the only experiment that has n=3. All other experiments contain 4-6 mice per experimental condition. We rationalized using both male and female mice because both human males and females have high lung cancer rates.

**Reviewer #2 (Public Review):**
Summary:In this manuscript, Ghazi et reported that inhibition of KRASG12C signaling increases autophagy in KRASG12C-expressing lung cancer cells. Moreover, the combination of DCC 3116, a selective ULK1/2 inhibitor, plus sotorasib displays cooperative/synergistic suppression of human KRASG12C-driven lung cancer cell proliferation in vitro and tumor growth in vivo. Additionally, in genetically engineered mouse models of KRASG12C-driven NSCLC, inhibition of either KRASG12C or ULK1/2 decreases tumor burden and increases mouse survival. Additionally, this study found that LKB1 deficiency diminishes the sensitivity of KRASG12C/LKB1Null-driven lung cancer to the combination treatment, perhaps through the emergence of mixed adeno/squamous cell carcinomas and mucinous adenocarcinomas.Strengths:Both human cancer cells and mouse models were employed in this study to illustrate that inhibiting ULK1/2 could enhance the responsiveness of KRASG12C lung cancer to sotorasib. This research holds translational importance.

We thank Reviewer #2 (R2) for the generally favorable review of our manuscript, and also for the more detailed critique that identifies potential weaknesses in the research, which we address on a point-by-point basis below.

Weaknesses:Additional validation of certain data is necessary.(1) mCherry-EGFP-LC3 reporter was used to assess autophagy flux in Figure 1A. Please explain how autophagy status (high, medium, and low) was defined. It's also suggested to show WB of LC3 processing in different treatments as in Figure 1A at 48 hours.

We thank the reviewer for this comment and agree that a more thorough description of how autophagy status is assessed using the Fluorescent Autophagy Reporter (FAR) would benefit the readers of our manuscript. Cells engineered to express the FAR are analyzed by flow cytometry in which we defined autophagy status by gating viable (based Sytox Blue staining), DMSO-treated control cells into three bins based on the ratio of EGFP:mCherry fluorescence. We gate all live cells into the 33% highest EGFP-positive cells (autophagy low) and the 33% highest mCherry-positive cells (autophagy high), and therefore, the proportion in the middle is also approximately 33% and considered the medium autophagy status. Again, these gates are based entirely on the DMSO-treated control cells, and all other treatments within the experiment are compared to settings on these gates. In response to a specific manipulation (sotorasib, trametinib, DCC-3116 etc) we assess how the specific treatment changes the percentages of cells in each of the pre-specified gates to assess increased autophagy (decreased EGFP:mCherry ratio) or decreased autophagy (increased increased EGFP:mCherry ratio).

Although LC3 processing and/or the expression of p62SQSTM1 are used by others as markers of autophagy, there is much debate in the literature as to how reliable immunoblotting analysis of LC3 processing or p62SQSTM1 expression are as measures of autophagy. Certainly, in our hands, we find that the Fluorescent Autophagy Reporter is a much more sensitive measure of changes in autophagy in various different cancer cell lines as we have described in previous papers (Kinsey et al., PMID: 30833748, Truong et al., PMID: 32933997 and Silvis & Silva et al., PMID: 36719686). Furthermore, in the omnibus publication that describes techniques for measuring autophagy (Klionsky et al., PMID: 33634751) the use of the FAR (or similarly configured reporters) is regarded as the gold standard for measuring autophagy status in cells. We have amended the Materials & Methods section of our manuscript to better describe the use of the FAR in measuring autophagy.

(2) For Figures 1J, K, and L, please provide immunohistochemistry (IHC) images demonstrating RAS downstream signaling blockade by sotorasib and autophagy blockade by DCC 3116 in tumors.

We thank the reviewer for the comment and have probed the tumors from the xenograft experiments in Figures 1J, K, and L for pERK1/2 and p62SQSTM1 to determine the biochemical activity of sotorasib or DCC-3116, respectively and have provided representative images below. We observed the expected decrease in pERK and p62 signal after sotorasib treatment in all three xenografted cell lines. We did observe the expected accumulation of p62 in the DCC-3116 treated tumors from the NCI-H2122 and NCI-H358 cell lines. There appears to be no difference between the vehicle and DCC-3116 treated tumors in the NCI-H358 cell line-derived tumors as detected by IHC.

**Author response image 1. sa2fig1:** 

(3) Given that both DCC 3116 and ULK1K46N exhibit the ability to inhibit autophagy and synergize with sotorasib in inhibiting cell proliferation, in addition to demonstrating decreased levels of pATG13 via ELISA assay, please include Western blot analyses of LC3 or p62 to confirm the blockade of autophagy by DCC 3116 and ULK1K46N in Figure 1 & Figure 2.

We appreciate the reviewer's comment and have performed an immunoblot analysis of cells treated with DCC-3116 or expressing ULK1K46N and probed for p62SQSTM1 and LC3 expression. We did observe the expected accumulation of p62 SQSTM1 in NCI-H2122 (ULK1K46N) cells treated with 1ug/ml doxycycline to induce expression of ULK1K46N compared to DMSO treatment. Additionally, we treated the human cell lines from Figure 1 with sotorasib and/or DCC-3116 and tested for p62SQSTM1 expression after 48 hours of treatment. In the human cell lines NCI-H2122 and NCI-H358, there was a decrease in the p62 signal with increasing doses of sotorasib, as expected. There was no detectable change in p62 levels in the Calu-1 cells by immunoblot. For LC3-I/LC3-II, there was only one detectable band in the NCI-H2122 cells, which makes it difficult to interpret the results and further emphasizes why we use the fluorescent autophagy reporter which is more sensitive than immunoblotting. There is no detectable change in LC3-I/LC3-II in the Calu-1 cells treated with increasing doses of sotorasib, but the expected decrease in LC3-I is observed with sotorasib treatment in the NCI-H358 cells.

(4) Since adenocarcinomas, adenosquamous carcinomas (ASC), and mucinous adenocarcinomas were detected in KL lung tumors, please conduct immunohistochemistry (IHC) to detect these tumors, including markers such as p63, SOX2, Katrine 5.

We have included IHC analysis of the adenosquamous carcinomas for the markers p63, SOX2, and Keratin 5 from the KL mouse in Figure 3 and the ASC tumors in Supplemental Figure 4, and thank the reviewer for this excellent suggestion. The straining for these markers is below. Of note, we tried two different SOX2 antibodies (cell signaling technologies #14962 and cell signaling technologies # 3728) and could not detect any staining in any section.

**Author response image 3. sa2fig3:** 

(5) Please provide the sample size (n) for each treatment group in the survival study (Figure 4E). It appears that all mice were sacrificed for tumor burden analysis in Figure 4F. However, there doesn't seem to be a significant difference among the treatment groups in Figure 4F, which contrasts with the survival analysis in Figure 4E. It is suggested to increase the sample size in each treatment group to reduce variation.

We have updated Figure 4E to indicate sample size for each treatment group and thank the reviewer for this suggestion. Any mice that remained on study through the entire 8-week treatment regimen were sacrificed after the last day of treatment (Day 56). Figure 4F indicates analysis of total tumor burden in all mice that remained on treatment for the full 8 weeks and mice that reached euthanasia criteria before the end of the 8-week treatment. Therefore, it is important to note that the mice in Figure 4F were not all euthanized on the same day. There is no statistically significant difference between the 3 treatment groups (sotorasib, DCC-3116, combination). This may be due to a lower sample size as well as ending the treatment at 8 weeks as opposed to continuing the treatment for a longer period of time. Although we agree that increasing the sample size would benefit the study, due to how long the GEMM model experiments take (12-16 weeks of breeding, 6 weeks for the mice to reach adulthood, 10 weeks of tumor formation post-initiation, 8 weeks of treatment = ~40 weeks) we would respectfully submit that the analysis of additional mice is outside the scope of the current revised manuscript.

(6) In KP mice (Figure 5), it seems that a single treatment alone is sufficient to inhibit established KP lung tumor growth. Combination treatment does not further enhance anti-tumor efficacy. Therefore, this result doesn't support the conclusion generated from human cancer cell lines. Please discuss.

We thank the reviewer for this observation. Indeed, KP lung tumors were sensitive to single agent DCC-3116 treatment, which is reflected in the tumor burden analysis. This was somewhat surprising to us as we have not previously detected much anti-tumor activity using 4-amino-quinoloines (chloroquine or hydroxychloroquine) or other autophagy inhibitors. It should be noted however that the KRASG12C/TP53R175H NSCLC model has a very low tumor burden overall (~4% in vehicle-treated mice). Additionally, our microCT imager cannot detect AAH and small tumors at the settings/resolution used. Therefore, we were limited in our ability to detect small tumors or hyperplasia by microCT imaging. Although there was a decrease in overall tumor burden with single agent DCC-3116 treatment, we could not demonstrate using microCT imaging that KRASG12C/TP53R175H lung tumors were actually regressing with single agent DCC-3116 treatment. The larger tumors that were detected appeared to show a cytostatic effect (i.e. no or slow growth) with DCC-3116 monotherapy. This may reflect our inability to detect regression of AAH or small tumors with the microCT. In all human cell lines tested, the only cell line that responded to single agent DCC-3116 treatment was NCI-H358 cells, which do have a complete heterozygous loss of the *TRP53* gene and lack TP53 protein. However, other cells that also have a loss of expression of TP53 expression (Calu-1) are insensitive to single-agent DCC-3116 treatment. Due to the low mutational burden of the KP mouse model compared to human NSCLC cell lines driven by mutationally-activated KRASG12C and the loss of TP53 function, it is difficult to directly compare GEM models to the human cell line models. Most of the human cell lines have alterations in other genes that are not altered in the KP mouse model which could affect the sensitivity of treatment.

**Recommendations for the authors:**

**Reviewer #1 (Recommendations For The Authors):**
Minor comments:(1) Figure legends are currently not adequate - information about the number and nature of replicates, stats, and definitions of the labelling used for stats should be added throughout. In Figure 5B, only two lines of four are labelled with * or ns.

We thank the reviewer for this comment and have included more details in the figure legends that describe replicates, statistical analysis and definitions of labeling. We also note that the methods section has a detailed description of the statistical analysis used.

(2) What statistical test is performed on Figure 5E to get a p < 0.05 between the vehicle and DCC group?

We performed a one-way ANOVA for all statistical analyses with more than 2 experiential groups. We thank the reviewer for pointing out this typo. These data points (vehicle vs. DCC-3116) are not statistically significant, which has been revised in the figure.

(3) The manuscript figures would be improved by the use of a colourblind-friendly palette.

We have previously published multiple manuscripts using this color scheme for the fluorescent autophagy reporter experiments and chose to use red and green as the reporter uses EGFP and mCherry. We wanted to keep this color scheme consistent across our publications and would prefer not to change the colors. However, we agree with the reviewer that the data should be accessible to all people and, therefore, have updated these graphs to include slashes over the red color to ease in telling the differences between the red and green colors. Thank you to the reviewer for this excellent suggestion.

(4) The manuscript should be fully checked for mouse (sentence case) and human (caps) gene (italics) and protein (non-italics).

In this manuscript we are using the nomenclatures approved by the HUGO Gene Nomenclature Committee (see here) in which:

Human genes are written as *KRAS*, *TP53* etc i.e. *ITALICIZED CAPS*

Mouse genes are written as *Kras*, *Trp53* etc: i.e. *Italicized and sentence case*

Human and mouse proteins are written as KRAS, TP53 etc: i.e. NON-ITALICIZED CAPS

In response to the reviewer’s suggestion, we have gone through the manuscript to check for this and make any appropriate changes. Of note, we intentionally refer to the mouse protein changes as KRASG12C/LKB1null or KRASG12C/TP53R172H (capitalized), as this references the protein change and not the nucleotide change that occurs in the gene.

(5) Adenosquamous is the correct term for the disease. In parts, it's referred to as adeno/squamous or adeno-squamous. The abbreviation ADC is also defined many times.

Thank you to the reviewer for this comment. We have corrected the manuscript text to only use adenosquamous and only define ADC in the first instance.

(6) Line 434 - "as previously described" but no reference.Typos:(1) Line 117 – either(2) Line 314 – synergistic(3) Line 317 – therefore(4) Line 502 – medium

We thank the reviewer for pointing out these typos and have modified the text appropriately.

**Reviewer #2 (Recommendations For The Authors):**
(1) The statement on Page 4, Lines 119-120, lacks clarity: 'Furthermore, LKB1 silencing diminishes the sensitivity of KRASG12C/LKB1Null-driven lung cancer perhaps through the emergence of mixed adeno/squamous cell carcinomas and mucinous adenocarcinomas. It is unclear whether this refers to the sensitivity to the combination treatment or to the KRASc inhibitor alone.

We thank the reviewer for this comment and agree that the statement lacks clarity. The intent of this statement was to refer to both single agent sotorasib treatment as well as the combination with DCC-3116.

(2) Page 5 Line 147 "KRASG12X ". Please correct this typo.

We thank the reviewer for this comment, but this is not a typo. We intended for this line to state KRASG12X to refer to cell lines with any KRASG12 alteration, e.g KRASG12D, KRASG12C, KRASG12S, KRASG12R etc.

(3) The color of the dots in Figure 5B labeling does not match the dots in the graph.

For all bar graphs in the manuscript, the dots representing individual mice are black, and the bar itself is color-coded based on treatment type. The dots in Figure 5B follow this pattern and are intended to be this way.

(4) Figure 5C depicts lung weight rather than tumor growth, contrary to the text description "regression of pre-existing lung tumors was detected by microCT scanning (Figure 5C, Figure S5)".

Figure 5C does not depict lung weight but the percent body weight change in treated mice, described in the figure legend. We thank the reviewer for pointing this out because we referenced the wrong panel in the text. The figures referenced should be Figure 5B, Figure S5. We have corrected this in the text.